# From Acquisition to Presentation—The Potential of Semantics to Support the Safeguard of Cultural Heritage

Jean-Jacques Ponciano †[iD], Claire Prudhomme †[iD] and Frank Boochs *,†

i3mainz, Department of Technology, University of Applied Sciences, 55128 Mainz, Germany; jean-jacques.ponciano@hs-mainz.de (J.-J.P.); claire.prudhomme@hs-mainz.de (C.P.)
* Correspondence: frank.boochs@hs-mainz.de
† These authors contributed equally to this work.

**Abstract:** The signature of the 2019 Declaration of Cooperation on advancing the digitization of cultural heritage in Europe shows the important role that the 3D digitization process plays in the safeguard and sustainability of cultural heritage. The digitization also aims at sharing and presenting cultural heritage. However, the processing steps of data acquisition to its presentation requires an interdisciplinary collaboration, where understanding and collaborative work is difficult due to the presence of different expert knowledge involved. This study proposes an end-to-end method from the cultural data acquisition to its presentation thanks to explicit semantics representing the different fields of expert knowledge intervening in this process. This method is composed of three knowledge-based processing steps: (i) a recommendation process of acquisition technology to support cultural data acquisition; (ii) an object recognition process to structure the unstructured acquired data; and (iii) an enrichment process based on Linked Open Data to document cultural objects with further information, such as geospatial, cultural, and historical information. The proposed method was applied in two case studies concerning the watermills of Ephesos terrace house 2 and the first Sacro Monte chapel in Varallo. These application cases show the proposed method's ability to recognize and document digitized cultural objects in different contexts thanks to the semantics.

**Keywords:** cultural heritage; point cloud; object recognition; linked open data; semantic enrichment; knowledge model; 3D acquisition

## 1. Introduction

Since the world heritage convention in 1972, UNESCO has worked actively to protect endangered world heritage sites and objects. It concerns cultural heritage that is subject to serious deterioration; significant loss of historical authenticity; loss of cultural significance; and the threat of human planning, armed conflict, or environmental factors (e.g., climatic and geologic). Actions are taken to protect cultural heritage by avoiding and mitigating threats and deterioration wherever possible. However, it is difficult to face the time and its consequences on cultural heritage. Therefore, its digitization and digital preservation are an opportunity to conserve it and share it with the public and between different organizations. The digitization process aims at converting information into a digital format and results in a digital representation. To preserve digital representation, it is necessary to ensure continued access to digital materials for as long as required. In 2005, Europe initiated the creation of a common access point to Europe's cultural heritage. Since this initiative, several European projects, such as the Europeana initiative (https://pro.europeana.eu/about-us/mission, accessed on 18 March 2021), ITN-DCH (https://www.itn-dch.net/, accessed on 18 March 2021), VIMM (https://www.vi-mm.eu/vimm-results/, accessed on 18 March 2021), and Dariah-EU (https://www.dariah.eu/about/mission-vision/, accessed on 18 March 2021), have been implemented to enable digital representation and the sharing of projects by making them reusable, visible, and sustainable. These projects promote cultural heritage documentation as Linked Open Data using semantic technologies. Linked Open Data and

semantic technologies facilitate the sharing of data and face digital format evolution and change over time that threaten digital preservation sustainability. However, the process from the digital acquisition to cultural heritage presentation is a long and challenging path, where research work is still necessary to improve it. This interest has been reinforced mainly for 3D digitization by the signing of the 2019 Declaration of Cooperation on advancing the digitization of cultural heritage in Europe. This study presents the potential of semantics to facilitate the process from 3D cultural heritage data acquisition to its presentation and thus support the safeguard of cultural heritage. The semantic technologies allow the meaning that is implicitly contained in data to be explicitly described (in logical form). This explicit meaning expressed through semantics enables machines and people to understand, share, and reason with one another. The proposed method aims at improving the process from 3D cultural heritage data acquisition to its presentation by providing an end-to-end process guided by expert knowledge through the use of semantic techologies. Challenges and motivations related to such a method are detailed in Section 1.1, and the related work is presented in Section 1.2. The method and its used knowledge model are described in Section 2. This method is composed of three steps:

- Data acquisition guided by a recommendation system for acquisition technologies.
- Data processing and structuring through knowledge-guided object recognition.
- Data presentation with cultural information thanks to an enrichment process from Linked Open Data.

The proposed method was applied in two case studies presented in Section 3. The first case study concerns the archaeological site of the terrace house 2 in Ephesos (Turkey). The second case study corresponds to the first chapel of the Sacro Monte in Varallo (Italy). Results of the proposed method applied to these case studies are presented in Section 4. Results obtained by the proposed method are discussed in Section 5, to conclude on the method Section 6.

### 1.1. Challenges

The process from 3D cultural heritage data acquisition to its presentation is composed of four main steps:

1. Data acquisition, which allows the digitization of a cultural object and produces unstructured data;
2. Data processing, which produces a structured data thanks to the segmentation, classification, and analysis of unstructured data;
3. Data enrichment, which consists of enriching the structured data with cultural heritage information and knowledge related to the structured data;
4. Data presentation, which allows the visualization of the structured and enriched data.

Each of these steps requires expert knowledge. The data acquisition of cultural heritage requires knowledge on acquisition technologies and cultural heritage to choose the most adapted technology according to the cultural object to acquire and the context [1–3]. Data processing requires computer scientist knowledge to define the most adapted processing according to the data and cultural objects or elements to recognize. The data enrichment requires cultural heritage and historical knowledge to add metadata and information related to the digitized cultural objects. All of these requirements show that this process is an interdisciplinary one.

In addition to the challenges specific to each stage of the process, this process's interdisciplinarity is a challenge that makes the process long and difficult. Providing an efficient process would require collaborative work between experts from different domains. However, understanding between the different experts, which is necessary to collaborate, is a difficult task that produces a sequence of isolated tasks rather than a continuous collaborative process. Such a process based on isolated and independent tasks is thus a long process that lacks a common pursued goal. A common goal would allow the optimization of each step according to the pursued final goal. This study proposes a

method to facilitate and improve the process from data acquisition to its presentation by using explicit knowledge representation. The knowledge representation aims to gather knowledge from the different experts and use it to guide users and the whole process according to a common goal. This goal is the presentation of enriched and structured cultural heritage data.

*1.2. Related Work*

The usage of semantics is not new in cultural heritage disciplines. They are commonly used to define standards for meta-, para-, and provenance information for documenting and archiving. Examples of such standards are LIDO [4] and MIDAS Heritage [5]. These XML schema standards are still used in cultural heritage. In recent years, however, the emergence of the Semantic Web has provided the much-required boost to semantic frameworks and technologies [6]. It also dictates how semantics are defined and used today. Techniques and tools that formalize semantics through formalized knowledge representations have become the norm in different fields applying semantics. Ontologies expressed through Web Ontology Language (OWL) [7] have evolved as major computational artefacts to provide logical representations of any particular domain of interest [8]. CIDOC-CRM is the most prominent and widely used ontology within the cultural heritage community [9]. In 2006, it became an ISO standard for publishing cultural heritage. Although semantics are commonly used for documenting and archiving cultural heritage, it is not often used to guide data processing and enrich data from Linked Open Data, which are other strengths of semantics that can be applied to the cultural heritage domain. Therefore, this section presents works related to approaches for data acquisition and processing, and then, works related to collect data and gather cultural heritage information thanks to semantics.

Although reviews comparing acquisition technologies for different application domains exist, such as [10,11], systems that guide users for data acquisition are rare. Cultural heritage objects are diversified (e.g., archaeological sites, heritage buildings, and paintings) with specific characteristics, documentation requirements, acquisition context, and application fields (e.g., preservation, restoration, and documentation). The acquisition techniques and technologies vary according to the application field and related cultural heritage objects to acquire. Therefore, the European project COST Action TD1201: " Colour and Space in Cultural Heritage (COSCH)" [12] has addressed the issue of determining preferred technical solution(s) according to data requirements needed to guide non-technical humanities experts. The approach presented in [13] proposes the COSCH-KR or COSCH Knowledge Representation, an ontology model to solve this issue. The ontology knowledge model constitutes interrelated semantics from technical and humanities domains involved in the optical recording of physical, cultural heritage assets. Inbuilt semantic rules infer the necessity of technical solution(s). COSCH-KR further applies semantics to the processing of these generated data as required by a cultural heritage application. Concerning cultural heritage data processing, the generation of annotated 3D models is nowadays widespread within the heritage community to disseminate and share information of cultural heritage objects. Various methodologies and algorithms have been applied to generate such computer-based 3D models. A review [14] presents the most popular methodologies and algorithms used to segment and classify 3D point clouds for the geospatial and heritage community. These authors highlight the advances made in this domain through the use of machine learning methods. Machine learning methods belong to the family of data-driven approaches. The main algorithms used to achieve machine learning are Markov Random Fields (MRF) (e.g., [15]) and quadratic programming [16], but also Associative Markov networks (AMN) [17]. Other approaches, such as [18–21], use deep learning techniques based on convolutional neural networks (CNN). A review [22] presents the different categories of these approaches. The limit of these machine learning and deep learning methods is the requirement of large data sets to obtain a satisfying result. Among the data-driven approaches, other popular methods are stochastic methods. Stochastic methods aim at the recognition of the context or are based on shape. The

recognition of context can provide semantic information describing a scene [23] or the geometry [24]. Shape-based recognition is used in [25] to identify semantic geometric classes by taking advantage of pre-structured knowledge. Ontologies are increasingly used to represent this knowledge and semantic information, all the more as they facilitate information retrieval [26] through the semantic web, and semantic techniques for querying cultural heritage data [27]. Through this work, the semantic technique is presented as being used to represent the result, but the semantic technique can also be used during recognition. The interest in using ontologies to process the data is mainly visible in the domain of image processing. In [28], a domain ontology is used to develop a recognition method. In [29], the detection and classification of objects are performed using ontology and reasoning techniques. However, most of these works only use semantic techniques to achieve some steps of the processing. The Knowledge-based object Detection in Image and Point cloud approach (KnowDIP) [30] uses semantic techniques at each step of the processing and is thus able to benefit from all advantages provided by the semantic technique, to both guide the process of computer-based modelling (through an adaptive selection of algorithms and an iterative classification) and represent the result of the 3D model understanding [31].

Semantics have an essential role in disseminating and sharing cultural heritage data collection and information gathering. Its main benefit is to solve problems of interoperability. It can enrich and homogenise the scheme of cultural heritage metadata to improve the searching and navigation functionalities of a cultural portal, as presented in [32]. It can also be used to publish and connect different sources of cultural data. Some approaches, such as [33], can connect a range of cultural heritage types, such as paintings, archaeological sites, archaeological exhibits and points of interest located in contemporary urban space. Such a connection and mapping are achieved through the CrossCult knowledge base's semantic-based design that aims to enhance the capabilities of the CrossCult platform and mobile applications. The proposed knowledge base contains an upper-level ontology based on CIDOC-CRM concepts and some additional concepts, such as Reflective Topic. It also includes the CrossCult Classification schema incorporated into the upper-level ontology. This knowledge base aims to connect and map information and data from cultural heritage institutions based on four flagship pilot cases from eight locations across Europe. Other existing approaches publish more specific cultural heritage data types (e.g., biography, artworks, and cultural heritage buildings) as Linked Open Data. The authors of [34,35] create an Irish CH knowledge base based on CIDOC-CRM, whose knowledge is derived from the Dictionary of Irish biography and linked to DBpedia. The work presented in [36] proposes that open linked data from the data on artworks and authors of the web portal of the Russian Museum be published. The proposed method consists of transforming data into RDF using CIDOC-CRM vocabulary. It links the thesauri of the British Museum to the *SKOS: concept* and specific concepts of CIDOC-CRM. It finally interlinks and enriches the knowledge representation with DBpedia. This enrichment consists of adding information about authors (e.g., date of birth and death and artistic movement author belongs to) first and annotating with links to DBpedia resource unstructured text as artwork descriptions and author biographies. Concerning cultural heritage building data, they require gathering both BIM information and cultural information. The authors of [37,38] propose the ontology HBIM that integrates Getty vocabulary and IFCOWL to create a catalogue of cultural heritage buildings and architectural complexes. This study belongs to the INCEPTION project aiming to provide a catalogue to be able to visualize, update, exchange, and divulgate cultural heritage buildings and architectural complexes. The work presented in [39] proposes a 3D model that is fully interoperable and rich in its informative content, enabling the user to query a repository composed of semantically structured and rich HBIM data. Existing approaches use semantic representation, mainly based on CIDOC-CRM vocabulary, to publish open data, gather different data sources, and facilitate the search and navigation of cultural heritage. However, only a few of them [34–36] exploit the strength of existing Linked Open Data, such as DBpedia. The approach presented in this study

proposes the exploitation of the rich interlinking of Wikidata entities to gather and collect information from different sources of Linked Open Data.

This related work study shows a lack of end-to-end approaches supporting cultural heritage documentation from data acquisition to its presentation. However, it highlights the potential of semantic to support this process and presents relevant semantic-based approaches to support some steps of this process. It thus allows us to determine COSCH-KR and KnowDIP approaches as relevant in supporting data acquisition and processing, respectively. These two approaches bring support in different contexts of application, and each of them provides a part of the knowledge domain intervening in the end-to-end process from cultural heritage acquisition to its presentation. As far as cultural enrichment is concerned, we are not yet aware of a flexible approach adapted to different contexts. However, a review of the related work study allows us to observe that most Linked Open Data sources and enrichment approaches are based on the CIDOC-CRM ontology. This ontology is, therefore, unavoidable when publishing and sharing cultural heritage data, information, and knowledge.

## 2. Method

This study proposes a method to support the process of cultural heritage acquisition to its presentation based on semantics, illustrated in Figure 1. This process is composed of three main steps:

- Cultural heritage acquisition.
- Data processing to structure it by recognizing objects.
- Enrichment from Linked Open Data to share and present cultural heritage.

This section presents the knowledge modeling based on the three steps of the method and then explains the three steps.

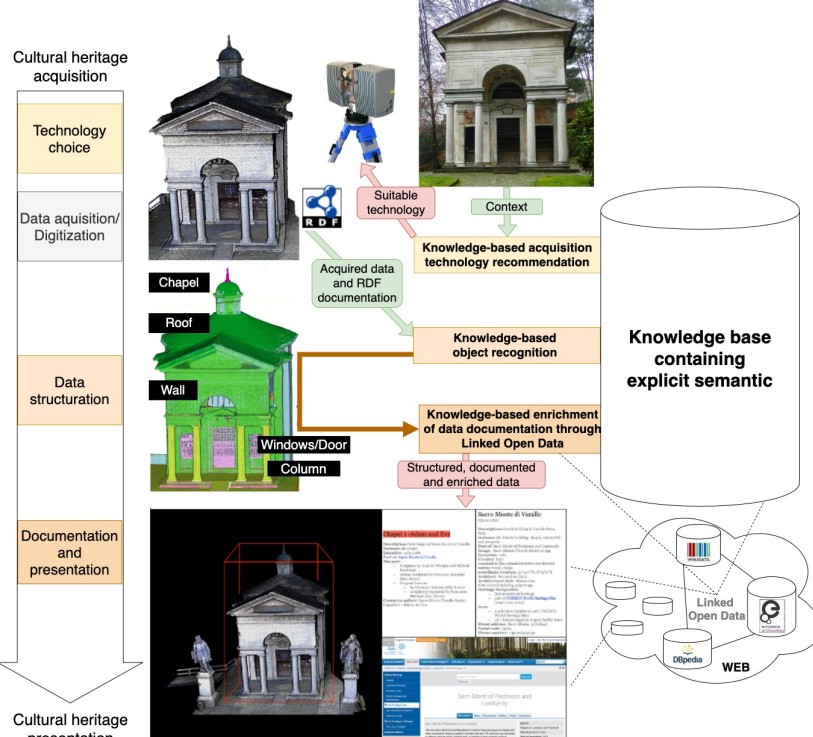

**Figure 1.** Proposed method overview. Green arrows correspond to user inputs, and red arrows correspond to automatically generated outputs.

## 2.1. Knowledge Modeling

The knowledge modeling approach aims at gathering three main domains of knowledge (acquisition, processing, and cultural heritage domains) to support the end-to-end process from cultural heritage acquisition to its presentation. Gathering these three domains of knowledge requires identifying the most suitable ontology for each of them and interlinking them. The related work presented previously has permitted the identification of such ontologies according to their usage. The ontology COSCH-KR contains vocabulary mainly related to the data acquisition process but also to data processing. It is used to guide non-technical humanities experts through the preferred technical solution(s) based on their data requirements. The ontology based on the KnowDIP approach contains mainly vocabulary for data processing and data acquisition. It is used to select and execute algorithms to provide the most adapted data processing according to its context of acquisition and objects to recognize. Finally, the CIDOC-CRM ontology is an unavoidable standard to represent cultural heritage knowledge.

Cultural objects are at the center of the end-to-end process of documentation. It is, thus, also at the center of knowledge modeling. However, it requires dissociating physical objects from virtual objects that result from digitization. Physical objects are physically existing objects or that have existed, whereas virtual objects are a representation of a physical object. Virtual objects correspond to digitized objects. By combining the two types of object with required knowledge domains, ontology concepts can be classified as belonging to:

1. Knowledge related to physical objects and their history;
2. Knowledge related to digitized objects and their associated process of digitization;
3. Knowledge related to the processing of digitized objects to recognize them and structure the data.

Figure 2 illustrates the main concepts of each ontology integrated into the knowledge base and their interlinking.

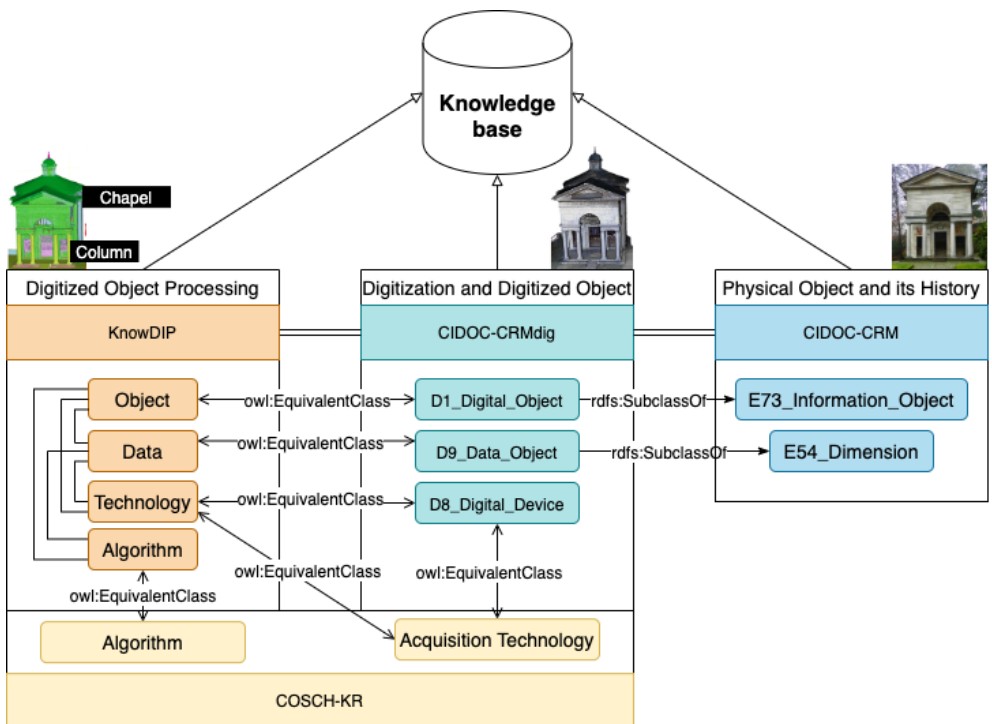

**Figure 2.** Knowledge base content.

Three or four (by considering an extension of CIDOC-CRM as one itself) ontologies have been integrated into the knowledge base. These ontologies are CIDOC-CRM,

CIDOC-CRMdig, COSCH-KR, and KnowDIP. CIDOC-CRM is a standard to document cultural heritage (as presented in Section 1.2). It allows us to describe physical objects associated with historical events, the place of events, and other cultural heritage information. CIDOC-CRMdig is an extension of CIDOC-CRM, specialized in representing the digitization process and resulting digitized objects. It allows the description of digitized objects associated with their digitization process and device. COSCH-KR or COSCH Knowledge Representation focuses on knowledge related to acquisition technologies and processing algorithms suitable for a cultural object type. KnowDIP vocabulary represents data processing knowledge. It describes mainly objects, data, acquisition technologies, and algorithms. These four main concepts are linked to allow the selection and execution of algorithms according to knowledge about objects to recognize and data to process.

The interlinking between these ontologies is mainly achieved by the property *rdfs: SubClassOf* that links concepts from CIDOC-CRMdig and CIDOC-CRM and by the property *owl: EquivalentClass* between concepts from CIDOC-CRMdig, COSCH-KR, and KnowDIP.

### 2.2. Recommendation of Acquisition Technologies

The COSCH approach uses rule-based reasoning on the COSCH-KR to recommend preferred acquisition technologies based on data requirements. It is addressed to non-technical humanities expert users. Inbuilt semantic rules infer the necessity of technical solution(s). COSCH-KR further applies semantics to the processing of these generated data as required by a cultural heritage application. The rule-based process of the recommendation of this approach is composed of three main steps.

1. The first step consists of determining the requirement of data for an optimal input according to a user's application context.
2. The second step consists of determining the most suitable acquisition technologies according to data requirements and characteristics of objects targeted by the user application.
3. The third step consists of computing parameters according to data and objects feature.

### 2.3. Object Recognition

The review of related work shows the value of combining the COSCH-KR and KnowDIP approaches to drive numerical algorithmic processing with semantics to apply the optimal algorithmic sequence and its parameters to the data. This combination provides flexibility to the algorithms, which are static, by adapting their parameterization and sequencing to the requirements of the situation. The KnowDIP approach uses the characteristics of the data (such as noise, roughness, and density) and objects (such as geometry and dimension) to make these adaptations. Furthermore, the semantics of one algorithm are analysed and correlated with other algorithms. This inter-algorithmic knowledge helps algorithms to align with the requirements, preconditions, and results of other dependent algorithms. Finally, a sequence of algorithms with the proper parameters is forged. This sequence considers not only the context in which they have to be executed but also the conditional restrictions of the other dependent algorithms [40]. This sequence of algorithms can be considered optimal for detecting and classifying objects in a particular dataset, as it takes into account each underlying constraint of the different situational and algorithmic consequences [41].

The application of semantics to the whole process provides the process with the necessary robustness to deal with unexpected situations arising during numerical processing. The recognition process is based on the KnowDIP ontology [30]. However, the KnowDIP ontology will require prior basic knowledge of the data, objects, and application requirements to trigger the algorithmic selection. COSCH-KR provides the initial basic knowledge to the KnowDIP ontology. This knowledge is derived from both asserted facts into the COSCH-KR and the inferred conclusion from these facts. The initial assertions through COSCH-KR pave the way forward for KnowDIP. KnowDIP further requires the definition of the scene and the anticipated objects inside, provided by the user (c.f. RDF documenta-

tion in Figure 1). This involves the detailing of these anticipated objects' characteristics, along with detailed knowledge of algorithms.

### 2.4. Object Enrichment with Cultural Heritage Information

The previous object recognition step provides the first level of data documentation with a label, a type, and geometrical information (as dimensions and shape) that characterize the cultural objects. However, it does not provide cultural and historical information related to these objects. Such a second level of documentation with related historical and cultural information is the enrichment process' goal. The enrichment process uses Linked Open Data (e.g., Wikidata (https://www.wikidata.org, accessed on 7 May 2021), DBpedia (https://www.dbpedia.org, accessed on 7 May 2021), and Europeana (https://www.europeana.eu/portal/en, accessed on 7 May 2021) [42], etc.) to retrieve knowledge and information related to the recognized cultural heritage objects.

Wikidata has the advantage of providing a large amount of information and being linked with both other Linked Open Data and Web data sources. Thus, by identifying the individual representing a recognized cultural object into Wikidata or its class, the process can link it to its equivalent representation inside the knowledge base. Thanks to this link, the process retrieves general and historical information about the object. The mapping between the knowledge base content and Wikidata is based on label matching. The process searches first for matching cultural heritage object instances labels. If the matching succeeds, the object instance will be linked to the matched instance in Wikidata through the property *owl:sameAs* (relationship between two individuals that indicates they are the same, https://www.w3.org/TR/owl-ref/#sameAs-def, accessed on 7 May 2021). Then, it aims to match the class labels of the cultural heritage object instances. In the case of matching, classes are linked through the property *owl:equivalentClass* (relationship between two classes that indicates they are equivalent, https://www.w3.org/TR/owl-ref/#equivalentClass-def, accessed on 7 May 2021). It finally aims for matching the label of individuals connected to a cultural object instance, by a property. The property *owl:sameAs* is added between two matched instances. After the mapping step, the process achieves the enrichment by integrating properties related to matched classes and individuals in Wikidata. The property enrichment is applied at a degree of 3 link levels. Thus, these steps retrieve different types of information specific to the cultural object instances, such as geospatial, cultural, and historical information, and add them to the knowledge base. The information and knowledge contained in the knowledge base are used to display the acquired point cloud with recognized objects and their associated enriched information in a Web interface.

The created Web interface is presented and explained in Figure 3. It is composed of four main parts. The first part (panel A in Figure 3) allows the display of the point cloud and the selection of the objects inside. The second part (panel B in Figure 3) always shows semantic information related to the selected object. The third part (panel C in Figure 3) displays further semantic information related to links clicked in the second part of the interface or links clicked directly in this part. This allows navigation among the knowledge base. The last part (panel D in Figure 3) allows other sources of information to be displayed, such as websites, images, and documents related to links clicked in the second or third part.

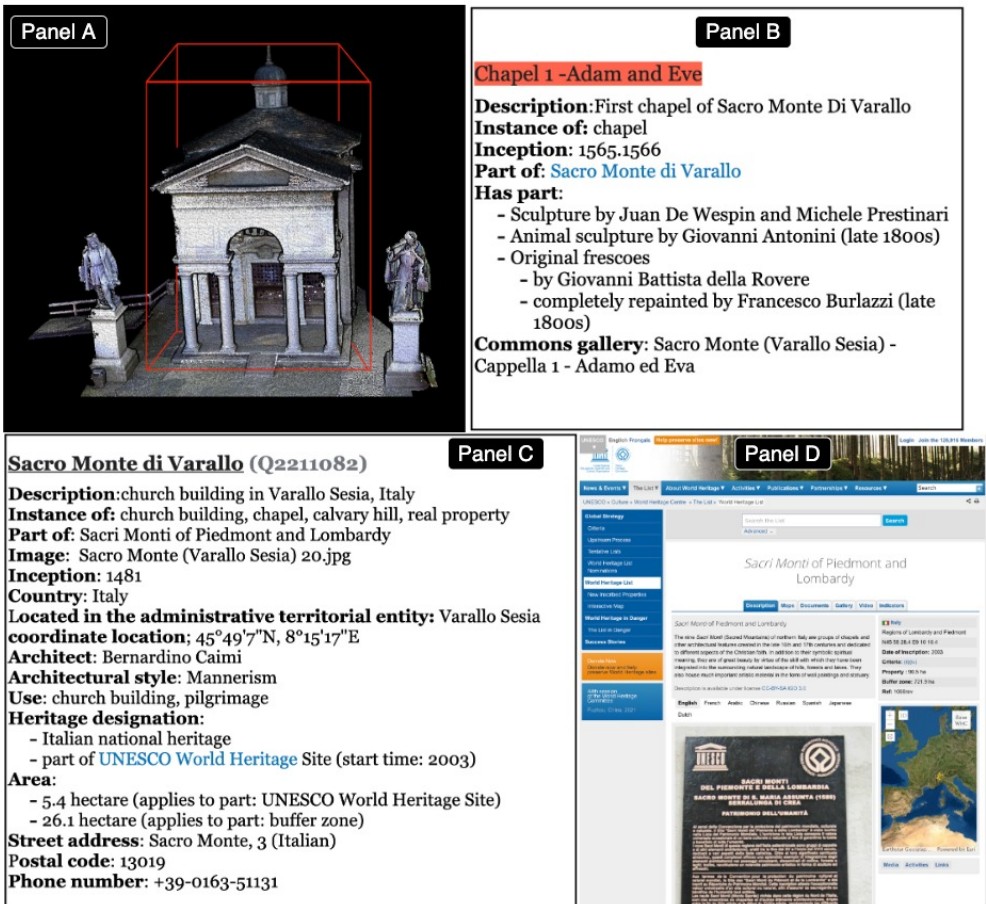

**Figure 3.** Explanation of interface made to present information and knowledge related to cultural heritage objects.

## 3. Materials

### 3.1. Case Studies

The proposed method for cultural heritage acquisition to its presentation was applied to two UNESCO World Heritage Sites. The first case study concerns the terrace house 2 of the Ephesos archaeological site in Turkey, and the second one concerns the first chapel of the Sacro Monte in Varallo, Italy. Both of them were digitized as a 3D point cloud.

#### 3.1.1. Terrace House 2 of Ephesos

The Ephesos ruin is composed of three main areas: the Byzantine city, late antique urban areas, and imperial urban areas. Terrace house 2, on which this paper focuses,

belongs to the imperial urban area. Terrace house 2 is an ancient ruin of about $100 \times 50$ m, excavated in the 1960s to 1980s, and partially 3D digitized in 2009/2010 (https://www3 .rgzm.de/ephesos/, accessed on 7 May 2021) [43]. Figure 4 illustrates the 3D digitized part of the terrace house 2. This 3D point cloud was acquired through the terrestrial laser scanner Leica HDS6000. It was then cleaned, registered, colored by combining images taken by an external reflex camera, denoised, and reduced. The point cloud is composed of 95,571,517 points. This digitized part is mainly composed of watermills and workshops, whose architectural structure components are in varying conditions: complete, incomplete, fragmented, and displaced.

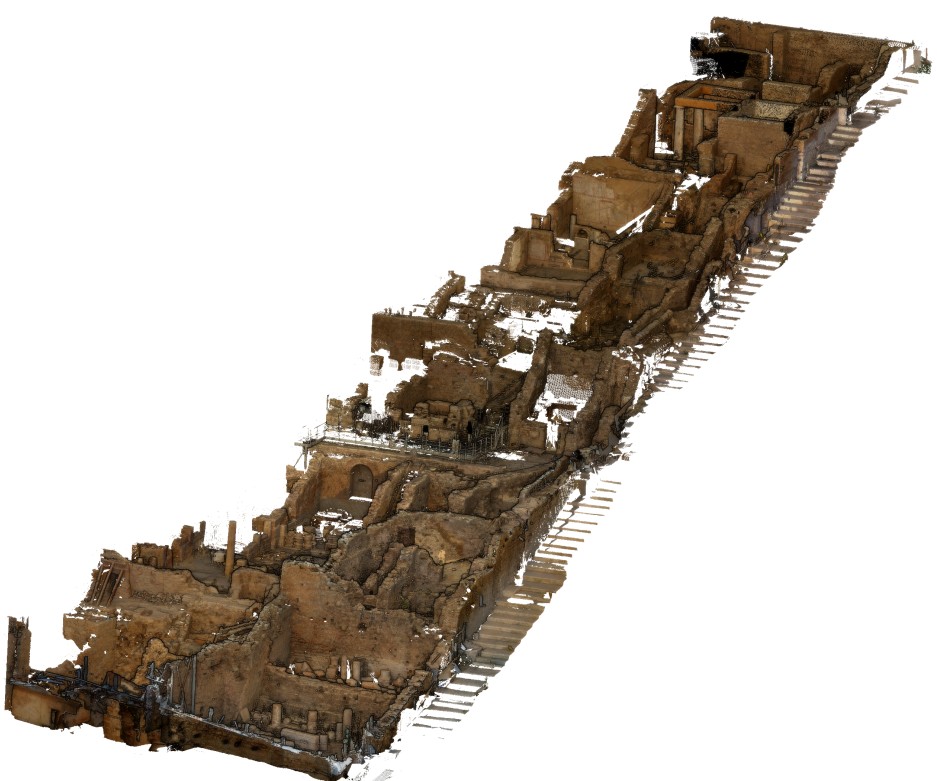

**Figure 4.** Original point cloud of Ephesos terrace house 2.

This case study focuses on recognizing and documenting the archaeological remains of watermills in terrace house 2. A watermill is composed of two rooms: one with a size of ca. $3 \times 4$ m that hosted the gearing mechanism and millstones in Late Antique/Early Byzantine times and a narrow room of ca. 0.60–1 m width corresponding to the waterwheel channel. In addition to providing the acquired data, the proposed method requires the user to provide information on its acquisition through RDF representation based on CIDOC-CRM and CIDOC-CRMdig ontology. The RDF representation for the acquisition of the watermills in Ephesos terrace house 2 is illustrated Figure 5.

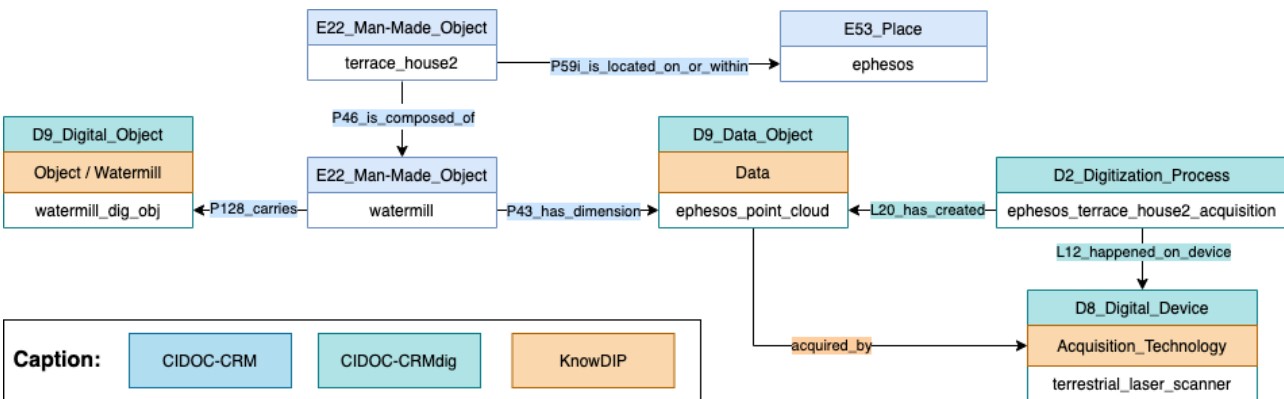

**Figure 5.** RDF representation of the watermills acquisition in Ephesos terrace house 2.

### 3.1.2. First Chapel of the Sacro Monte in Varallo, Italy

The Sacro Monte di Varallo is located in Valsesia, in the province of Vercelli. It is the oldest of the Sacred Mounts of the Piedmont and Lombardy regions. Its history began at the end of the 15th century when the Milanese Franciscan friar Bernardino Caimi decided to reproduce the holy places of Palestine in Valsesia on his return from the Holy Land. The "New Jerusalem", as the Sacro Monte was called, was initially intended to reproduce the distant sites of the Christian tradition for all those who could reach them. Inside these places, there were images, paintings, and sculptures that evoked the related events in the history of Christ's life. The Sacro Monte di Varallo consists of a basilica and forty-five frescoed chapels, populated by more than eight hundred statues. Many important Piedmontese artists have contributed to the decoration and completion of this extraordinary complex, including, in addition to Gaudenzio Ferrari, Bernardino Lanino, Tanzio da Varallo, Enrico's brothers, the Morazzones, Dionigi Bussola, and Benedetto Alfieri. This second case study focuses on the first chapel, "Adam and Eve", in front of which the Sacro Monte tour begins, which represents the original sin. Although this fact is not directly related to the life of Jesus, it is intended to explain the life of Christ by answering the theological question as to "why God became man in Christ". One of the answers is to save humanity from the sin present from the beginning, hence the Adam and Eve Chapel.

The first chapel was acquired outdoors through a terrestrial laser scanner (TLS), including a FARO Focus 3D X 130, 120, and a Riegl VZ-400 combined with a Unmanned Aerial Vehicle (UAV) equipped with a SONY Ilce 5100L, and a DJI Phantom 4 Pro with an integrated camera (more information on data acquisition is available at: http://archdataset. polito.it/dataset_specs/, accessed on 7 May 2021). The 3D point cloud representing the first chapel (c.f. Figure 6) is derived from an open dataset [44]. It is colorized and composed of 3,783,412 points.

Similarly to the previous case study, the user provides the RDF representation for the acquisition of the chapel "Adam and Eve" in the Sacro Monte Varallo. Figure 7 illustrates this RDF representation.

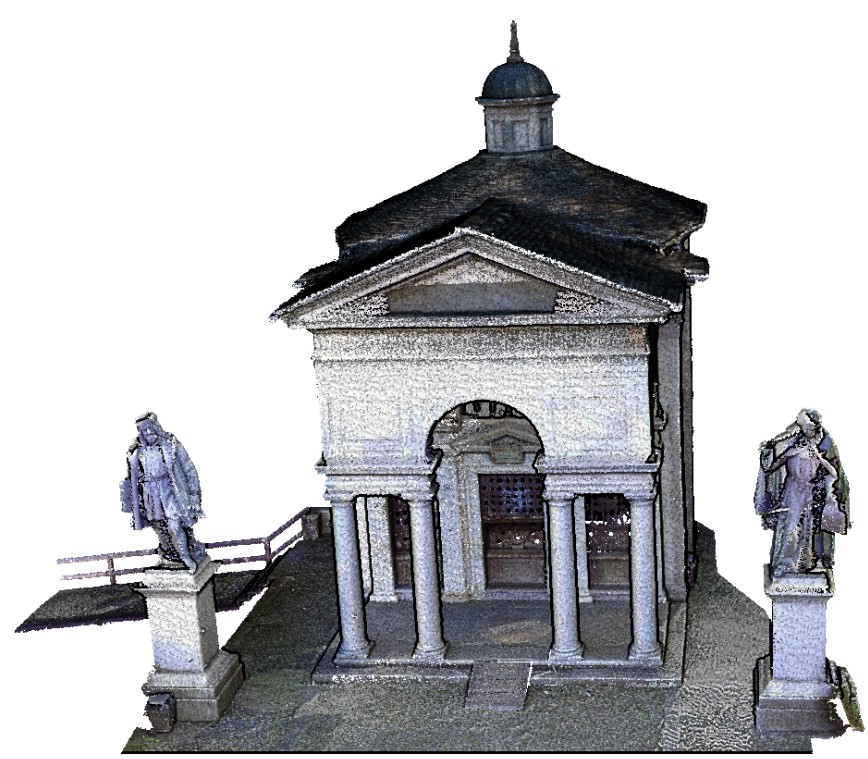

**Figure 6.** Original point cloud of the first chapel of Sacro Monte.

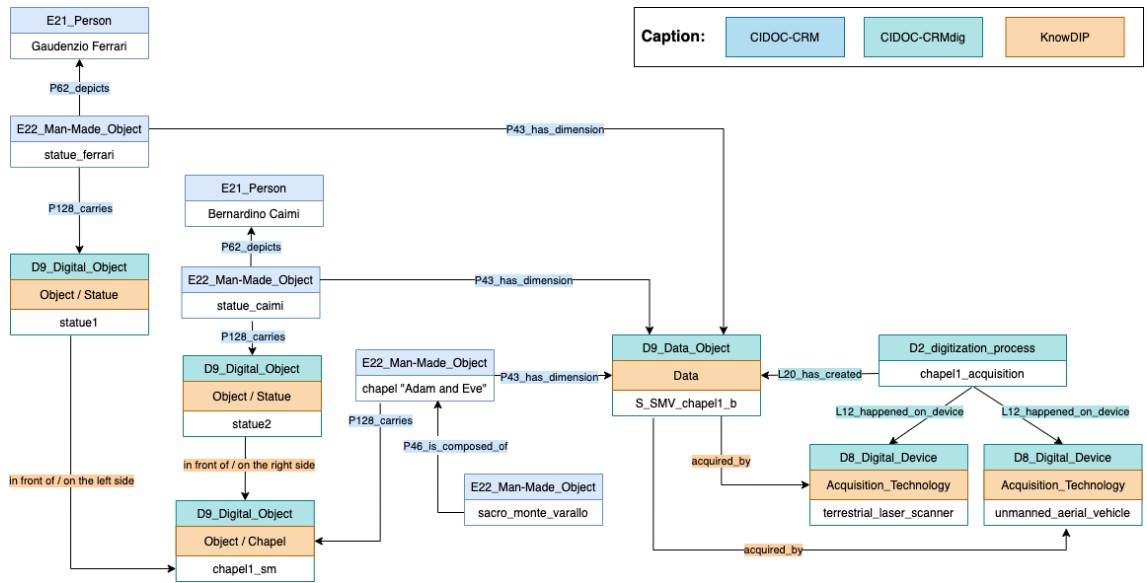

**Figure 7.** RDF representation of the chapel "Adam and Eve" acquisition in the Sacro Monte Varallo.

### 3.2. Object Modeling Related to the Case Studies

The method proposed in this paper is guided by semantics thanks to a knowledge base gathering experts' knowledge. The process of a case study requires integrating into the knowledge base a minimum of knowledge about the concerned objects. This section presents the integrated knowledge related to the two case studies.

#### 3.2.1. Watermill of the Ephesos Terrace House 2

The Ephesos terrace house 2 point cloud is a complex architectural structure of walls, floors, doors, windows, roofs, ceilings, and columns, which are in varying condition:

complete, incomplete, fragmented, and displaced. Only the ground floor is preserved; the roof and ceiling(s) are missing. For example, today, a formerly vertical wall is tilted, or a formerly vertically standing column has fallen and now lies on the floor. According to these data's archaeological knowledge, the watermill is semantically described in the ontology as composed of two specific rooms: a large room and a narrow room. A room is described as having at least three building walls and one connected floor. The large room is characterized by its size of ca. 3 × 4 m. The narrow room, characterized by a ca 0.60–1 m width, accompanies the large room. These two rooms share one wall. In addition to this information about the general geometric condition of a watermill and a room, more simple objects, such as walls and the floor, are described in the KnowDIP ontology. A complete wall is described as a rectangle shape with a vertical orientation and a width of ca. 0.6 m, and a floor as a rectangle shape with a horizontal orientation. Information about their topological relations is also represented: a wall is connected to at least one other wall and at least one floor. A floor is under the walls and can be connected with another floor. The complete object modeling is detailed in [45].

### 3.2.2. First Chapel of Sacro Monte

The case study of the first chapel of Sacro Monte is composed of three main objects: the chapel, two statues, and stairs. The chapel is described as a composition of other objects, which are the roof, wall, floor, column, arch, molding, vault, door/window, and stairs. Each object is described as a composition of other objects or through a geometry. They are characterized by different characteristics (such as colors, width, and height) and topological relations with other objects. Taking the example of a column description, illustrated in Figure 8, a column is described through the geometrical shape of a cylinder. It has a width of between 0.5 and 0.7 m and a height of between 3 and 3.5 m. The column description also contains a relation description with other columns: columns are aligned.

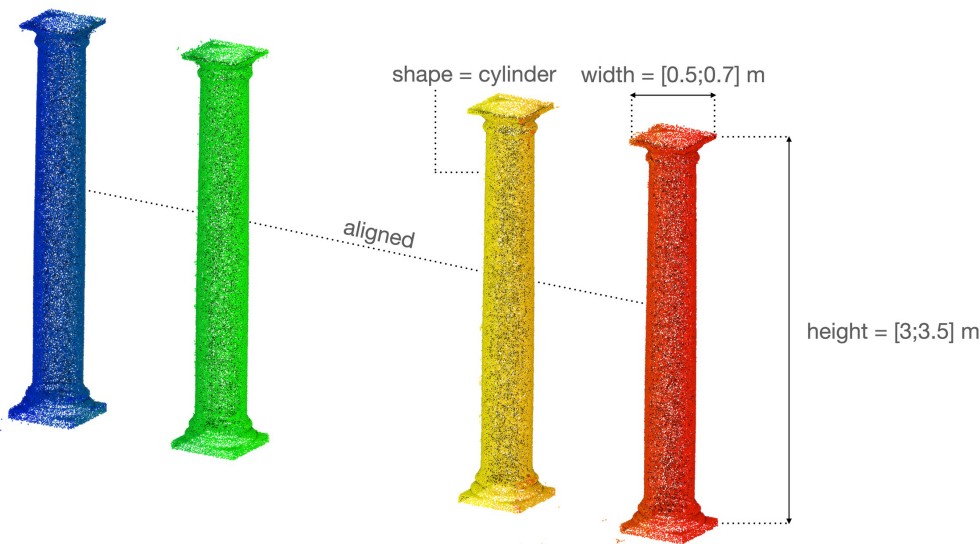

**Figure 8.** Knowledge about the object column.

## 4. Results

This section presents the results obtained for each step of the proposed method applied to the case studies.

### 4.1. Recommendation of Acquisition Technologies

The recommendation of the acquisition technologies was performed according to the application objective. From the user's application goal, COSCH-KR derives the characteristics of the acquired data and objects. These characteristics are added to the knowledge representation of the scene, objects, and data in the KnowDIP ontology. In the presented

case studies, the objective is "Detection and classification of cultural heritage site objects". From this input, COSCH-KR determines that 3D data should be "denoised and of medium to high quality" as the optimal input quality for such an application. Considering the size of the objects, it recommends Terrestrial Laser Scanner (TLS) as the acquisition technology. The recommended technology (TLS) provides hints on the characteristics of the data (such as quality level, density, noise, and alignment). The resulting data are, therefore, 3D point clouds of medium to high quality. These inferred findings provide preliminary knowledge about the objects and data. For example, the preliminary knowledge of the objects allows the approach to determine that the roughness of the objects is high. This inference allows KnowDIP to determine the tolerance value required for the calculation of point orientation (normal) during segmentation.

*4.2. Object Recognition Results*

The recognition process focuses on watermills in the first case study and on chapel components and statues in the second case study.

### 4.2.1. Result of Watermill Recognition in Ephesos Terrace House 2

Object recognition is based on the semantic descriptions of objects in the knowledge base. In the first step, the process automatically establishes a dependency graph to determine which parts or objects should be recognized in priority, as explained in [46,47].

For the application case of watermill detection, a watermill is defined in the knowledge base as being composed of two rooms: one large and one narrow. Thus, the strategy to recognize the watermills consists of first recognizing the different rooms present in the application case. The detection of rooms is explained in [45] and consists of identifying the floors connected to at least two walls to form a room. The detection of walls and floors is explained in [41], and consists of applying a segmentation mainly based on point orientation, then extracting the segments' geometrical characteristics to classify them according to these characteristics. Figure 9a illustrates the recognition of rooms in the point cloud (a random color is assigned to each room).

The historical knowledge on the watermills allows the precise description of the dimensions of the room which constitute the mill. The dimensions of each room are then extracted to identify the large room and the narrow room. When a narrow room is connected to a large room, these two parts are combined to identify a watermill. Figure 9b shows the watermills' recognition in the point cloud, with their two rooms (the narrow room in blue, and the large room in red).

### 4.2.2. Results of the Sacro Monte First Chapel

Object recognition requires a segmentation step to group points into homogeneous segments. This segmentation is based on automatic segmentation processes (such as [48]) and region growing algorithms (such as [49]). Figure 10 (in the center) illustrates the result of this step on the Sacro Monte first chapel by assigning a unique color to each segment.

Geometric characteristics of these homogeneous segments, such as density, dimensions, planarity, and orientation, are then extracted. The extraction of these characteristics is performed by data processing algorithms, such as the following:

- GetHeight: characterizes the height of a segment.
- GetWidth: characterizes the width of a segment.
- GetLength: characterizes the length of a segment.
- GetVolume: characterizes the volume of a segment.
- GetArea: characterizes the area of a segment.
- GetLocation: characterizes the location of a segment.
- GetMeanNormal: characterizes the mean normal of a segment.
- GetPointCount: characterizes the point number of a segment.
- GetMeanColor: characterizes the mean color of a segment.
- GetResolution: characterizes the resolution of a segment.

- GetDistance: characterizes the Euclidean distance between two segments.
- GetAltitude: characterizes the altitude of a segment.
- GetDistance: characterizes the Euclidean distance between two segments.
- isParallele: characterizes the parallelism between two segments.
- isBetween: characterizes segments for which their position is between two other segments.
- isAligned: characterizes segments that align with other segments.

These characteristics provide clues about the objects or portions of objects to be recognized. These clues are then integrated into the knowledge base to perform reasoning that consists of classifying and merging sets according to their characteristics and logical descriptions of objects or portions of objects. The result of this classification applied to the first chapel of Sacro Monte is illustrated in Figure 10 (on the right).

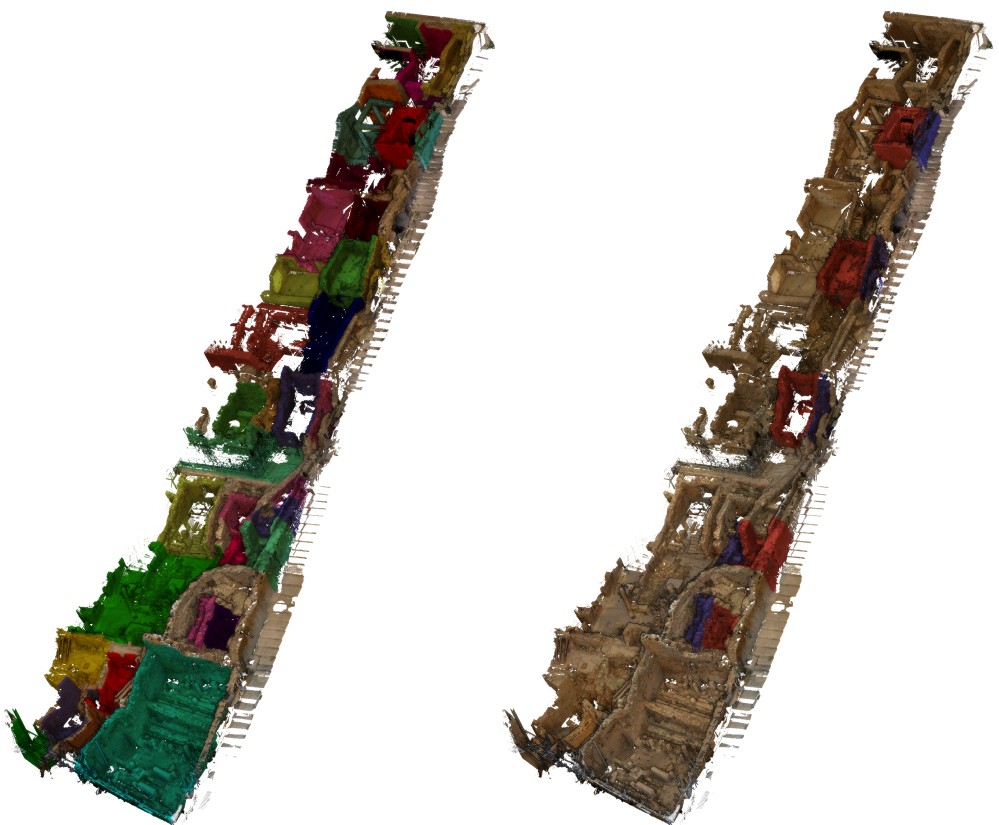

**Figure 9.** Object recognition results in Ephesos terrace house 2 point cloud: (**a**) room recognition and (**b**) watermill recognition.

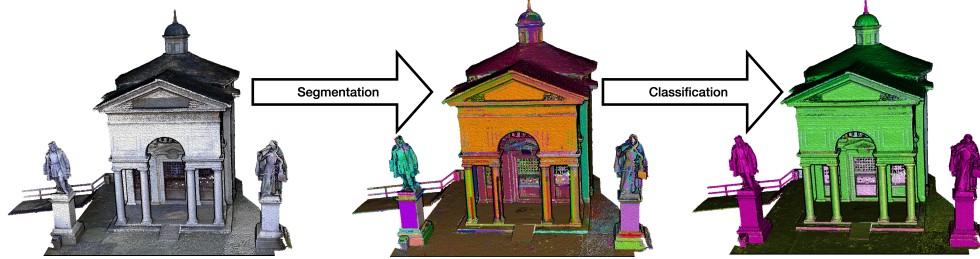

**Figure 10.** Results of object recognition phases (segmentation and classification) applied to the chapel of the Sacro Monte.

Object recognition based on geometric characteristics provides a first object recognition result that determines the location of the different elements. For example, the columns shown in yellow in Figure 11 on the left are slightly detected. Only a few portions are

recognized, but these portions locate the columns. Moreover, the historical knowledge of this application case describes the presence of four columns and two statues resting on a base. Due to the localization of portions of columns, the process can segment the columns to recognize them more effectively. This new segmentation uses a bounding box, whose dimensions correspond with those described by the column's knowledge. This further segmentation improves the recognition process, as shown in Figure 11 for the columns (in yellow).

Similarly, by combining the location of recognized objects with historical knowledge, the proposed approach can improve statue recognition by segmenting the base and the statue and then classifying the segments. This improvement is visible in Figure 11 (on the right with the base in orange and the statue in red).

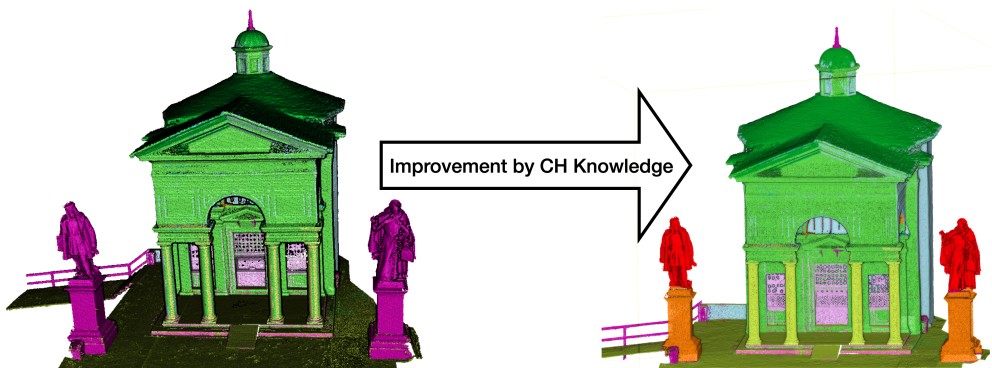

**Figure 11.** Results of the object recognition improved by cultural heritage (CH) knowledge.

An evaluation of the object recognition in this case study was performed by calculating four metrics (F1 score, IoU, precision, and recall) on the results of the semantic segmentation derived from the object recognition. This derivation considers the statue and its base as an "other" class to maintain consistency with the ground truth related to this application case. Table 1 presents the results of the four metrics for each object classified in the ground truth.

**Table 1.** Metric results of the object recognition for the Sacro Monte first chapel.

| Class (Index Code) | F1 Score | IoU | Precision | Recall |
|:---:|:---:|:---:|:---:|:---:|
| Column (1) | 0.981 | 0.963 | 0.999 | 0.963 |
| Stairs (6) | 0.382 | 0.236 | 0.779 | 0.253 |
| Floor (3) | 0.903 | 0.823 | 0.835 | 0.983 |
| Moldings (2) | 0.682 | 0.517 | 0.691 | 0.673 |
| Other (9) | 0.872 | 0.773 | 0.965 | 0.796 |
| Wall (5) | 0.867 | 0.765 | 0.846 | 0.888 |
| Door/Window (4) | 0.801 | 0.668 | 0.821 | 0.782 |
| Roof (8) | 0.880 | 0.786 | 0.875 | 0.885 |
| Vault (7) | 0.728 | 0.572 | 0.704 | 0.753 |
| Arch(0) | 0.341 | 0.205 | 0.492 | 0.261 |
| Total | 0.833 | 0.715 | 0.833 | 0.833 |

### 4.3. Object Enrichment Results

This subsection presents the results of semantic enrichment related to cultural objects recognized in the previous step for each case study.

### 4.3.1. Ephesos Terrace House 2

In the first case study, the watermill concept was initially mainly represented through its geometric aspect with KnowDIP vocabulary and a user's knowledge. Thanks to the process of enrichment, this concept was interlinked with its equivalent Wikidata (*wkd:Q185187*) and DBpedia (*dbo:Watermill*) concepts. It has also been enriched by properties associated

with its equivalent concepts from Wikidata and DBpedia. The property enrichment is applied at a level of three links, allowing the knowledge base to be enriched with individuals and concepts related directly to the watermill concept or to individuals and concepts related to it through a property. It thus allows adding instances of "Terrace houses (Ephesus)", "Terrace House 2 (Ephesus)", and "Ephesus" and also their associated properties from Wikidata. Figure 12 illustrates the enrichment related to the watermill concept.

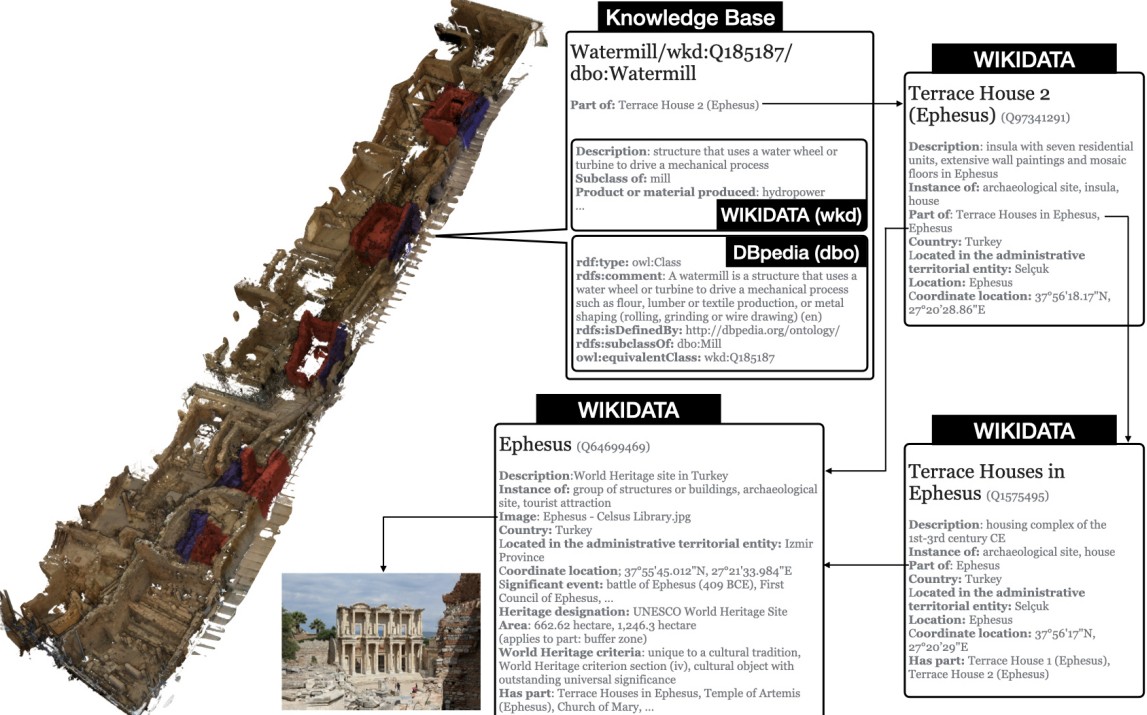

**Figure 12.** Enriched cultural heritage information related to watermills of Ephesos Terrace House 2.

### 4.3.2. First Chapel of Sacro Monte

The second case study is mainly composed of three cultural heritage objects: the chapel "Adam and Eve", the statue of Gaudenzio Ferrari, and the statue of Bernardino Caimi.

The enrichment process applied to the chapel instance results in matching the instance of "Sacro Monte di Varallo" (*wkd:Q2211082*), where is located the chapel. It then adds all properties related to the Sacro Monte instance in Wikidata, thus providing links to different web resources as an image of the Sacro Monte or the world heritage site of Unesco associated with the Sacro Monte. The matching step with the Italian chapel label identifies a commons gallery of Wikimedia related to it. This gallery provides different images related to the chapel of Adam and Eve. Figure 13 provides an overview of the enrichment related to the chapel.

The object recognition process first identified that the point cloud contains two statues. Then, thanks to the RDF representation provided by the user and the topological relations between the chapel and the statues, it identified the first one as representing Gaudenzio Ferrari, and the second one as representing Bernardino Caimi. During the enrichment process, the concept *Statue* was matched with the concept *wkd:Q179700* in Wikidata. The statue instance, representing Gaudenzio Ferrari, was not matched with an instance in Wikidata. However, the person of Gaudenzio Ferrari was matched with the instance *wkd:Q734108*. In addition to the information related to Gaudenzio Ferrari, the enrichment from its instance of Wikidata provided links to further semantic descriptions in Linked Open Data as from *dati.beniculturali.it* and Europeana but also web resources, such as a wikimedia commons gallery containing images from his artwork. Figure 14 illustrates

these different sources of information related to Gaudenzio Ferrari that have enriched the knowledge base.

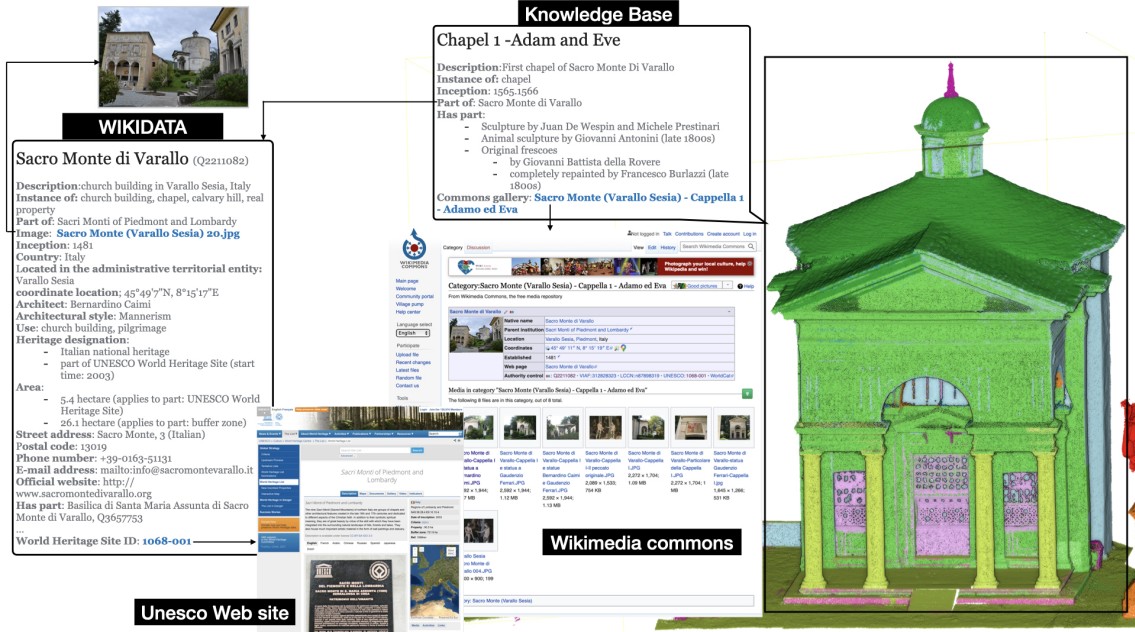

**Figure 13.** Enriched cultural heritage information related to the chapel.

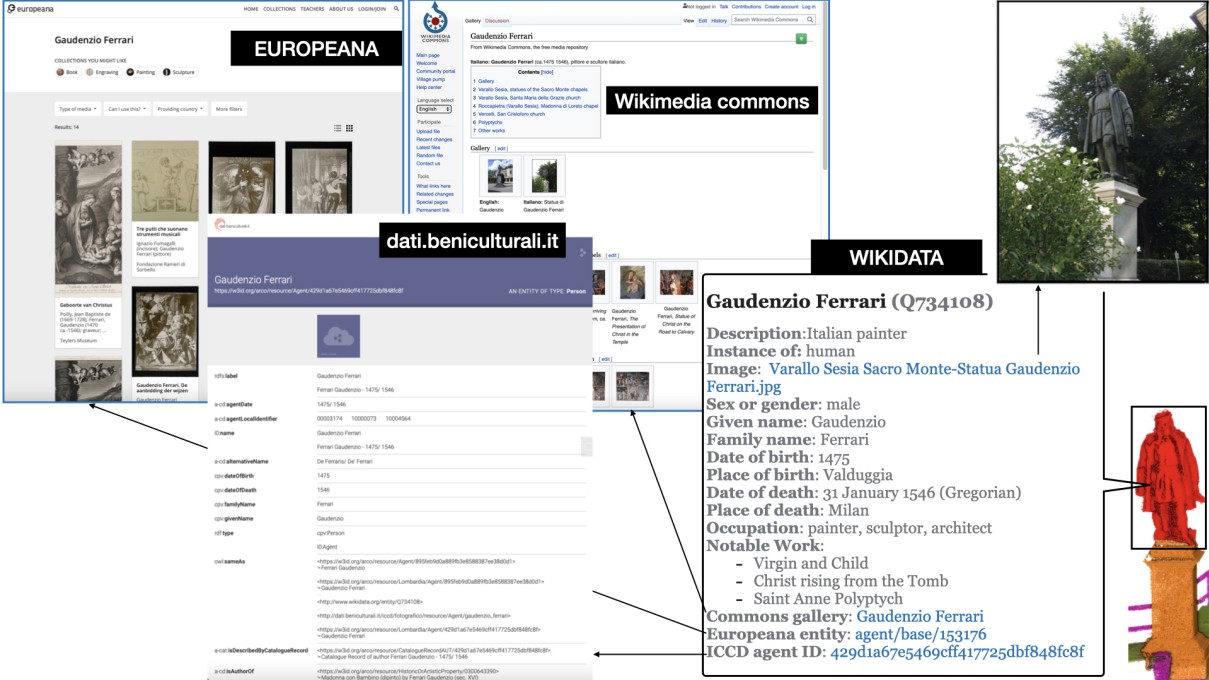

**Figure 14.** Enriched cultural heritage information related to first statue in front of the chapel.

Similarly to the first statue, the enrichment related to the second statue concerns the person that it represents. This process matched the instance of Bernardino Caimi with the instance *wkd:Q3638697* in Wikidata. From this matching, the process then added the related properties providing access to different web resources as an image of its statue or an Italian website describing its biography.

## 5. Discussion

This section discusses and explains the results obtained for the different steps of the proposed method.

### 5.1. Recommended Acquisition Technology Discussion

Comparing the acquisition technologies recommended by COSCH-KR with the acquisition technologies used by the experts in the two presented case studies, we see that in both case studies, a TLS was used as recommended by the approach.

### 5.2. Object Recognition Discussion

This subsection explains the object recognition results obtained for these two case studies and the role of the knowledge that guided and impacted these results. It thus discusses the strengths and weaknesses of the knowledge that was used to guide object recognition in such cases.

#### 5.2.1. Discussion on Watermill Recognition in Ephesos Terrace House 2

The terrace house 2 is composed of seven watermills, built during three construction phases [43], as illustrated in Figure 15 on the right. However, the acquired point cloud contains only six watermills, as illustrated by the water wheel outside of the point cloud at the middle bottom of Figure 15. Among these six watermills contained in the original point cloud, the proposed method recognized five of them, illustrated in Figure 15 on the left. By comparing the watermill construction phases and the result of watermill recognition, we can observe that four recognized watermills belong to the third construction phase. They are thus more recent than the unrecognized watermill belonging to the first construction phase.

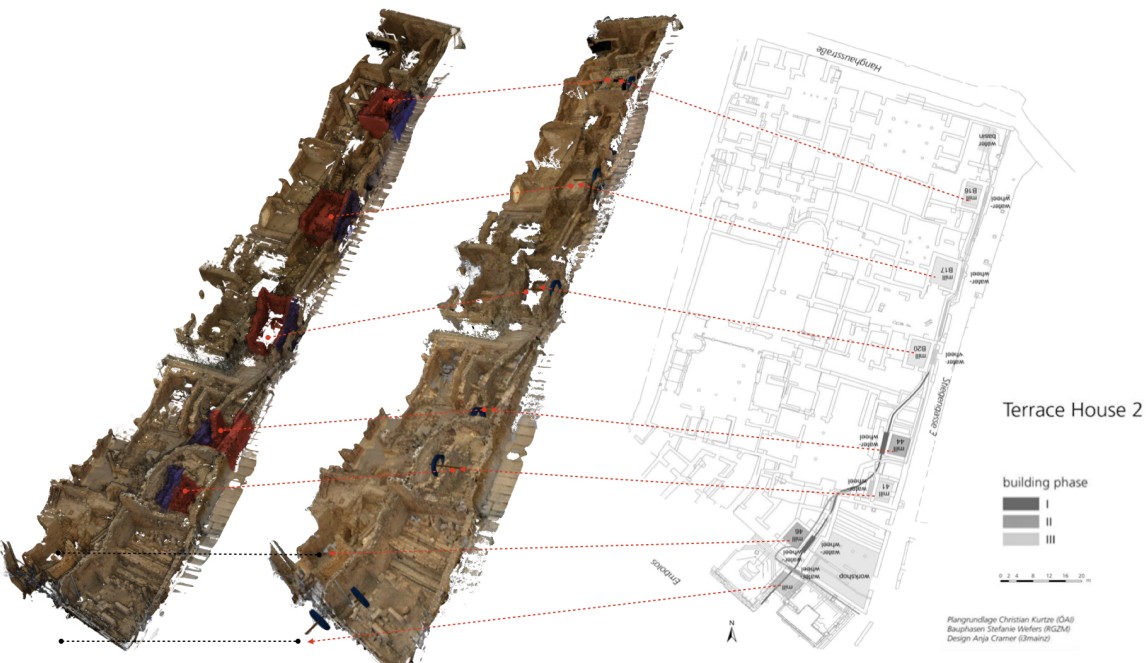

**Figure 15.** Watermill recognition in Ephesos terrace house 2: (**left**) point cloud with detected watermills in blue and red; (**middle**) watermills with their water wheel [50]; (**right**) terrace house 2 plan presenting watermills and their construction phases [50].

By observing in detail the point cloud illustrated in Figure 16, we can see that the narrow room (framed in yellow) of the unrecognized watermill is no longer present or was not acquired. This observation explains that the recognition process was not able to recognize it, as it no longer fits the watermill description that specifies the connection of two

rooms, such as the narrow room for the water wheel. The destruction of rooms can be due to time erosion but also other construction phases. This is the limit of object recognition: it can recognize cultural heritage objects that are partially destroyed, but not when a complete component is destroyed or not acquired. Only archaeological interpretation would be able to recognize such an object, which requires further knowledge specific to the archaeology domain.

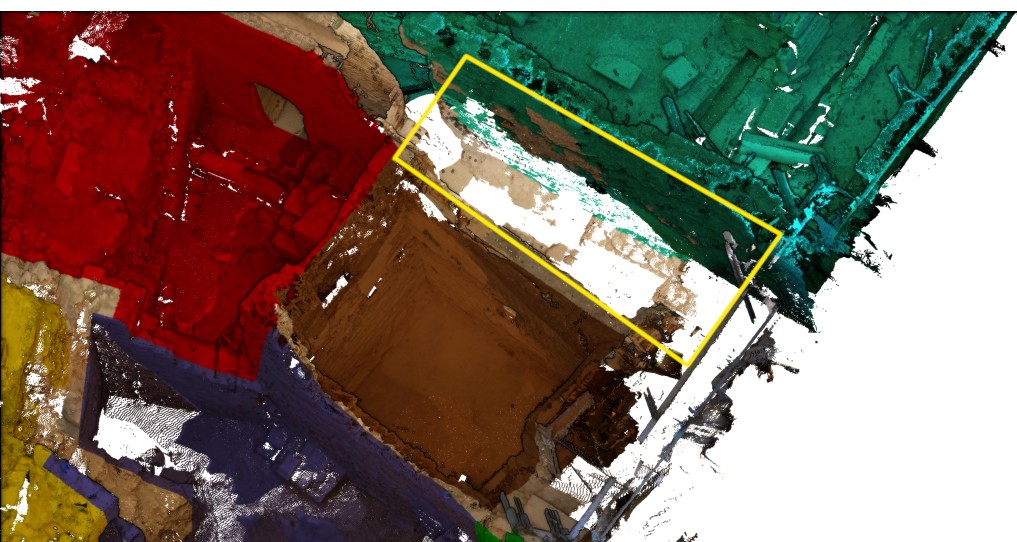

**Figure 16.** The unrecognized watermill composed of the large room (in brown) and the narrow room (framed in yellow).

### 5.2.2. Discussion on the First Chapel of Sacro Monte

The recognition of objects in the chapel provides a basis for locating and identifying different portions of the point cloud. Although it is not perfect, its quality is comparable to that obtained by the literature's best classification approaches. The quality of the detection depends mainly on the quality of the knowledge that describes the objects that need to be recognized. In this case study, the columns' knowledge is sufficiently accurate and diversified to allow a detection process with an accuracy of 99.9%. On the contrary, if the objects' knowledge is not sufficient, such as that for the arch and stairs, then the objects are less recognized (e.g., only 25% of the arch and stairs are recognized). However, this knowledge can be enriched in a supervised way by experts or automatically with approaches such as [40,51], which use a self-learning process based on ontologies to enrich the knowledge.

Figure 17 illustrates the optimal result that can be obtained for the object recognition of the Sacro Monte chapel with supervised knowledge enrichment.

The optimal result is obtained by using the knowledge of all the object categories that compose the point cloud to force the points' classification into one of these classes. Figure 18 shows an isolated view of the recognized objects for each of these categories (arch, floor, spotlight, statue base, bin, gutter, roof, vault, column, molding, stairs, wall, fence, needle, statue, and window).

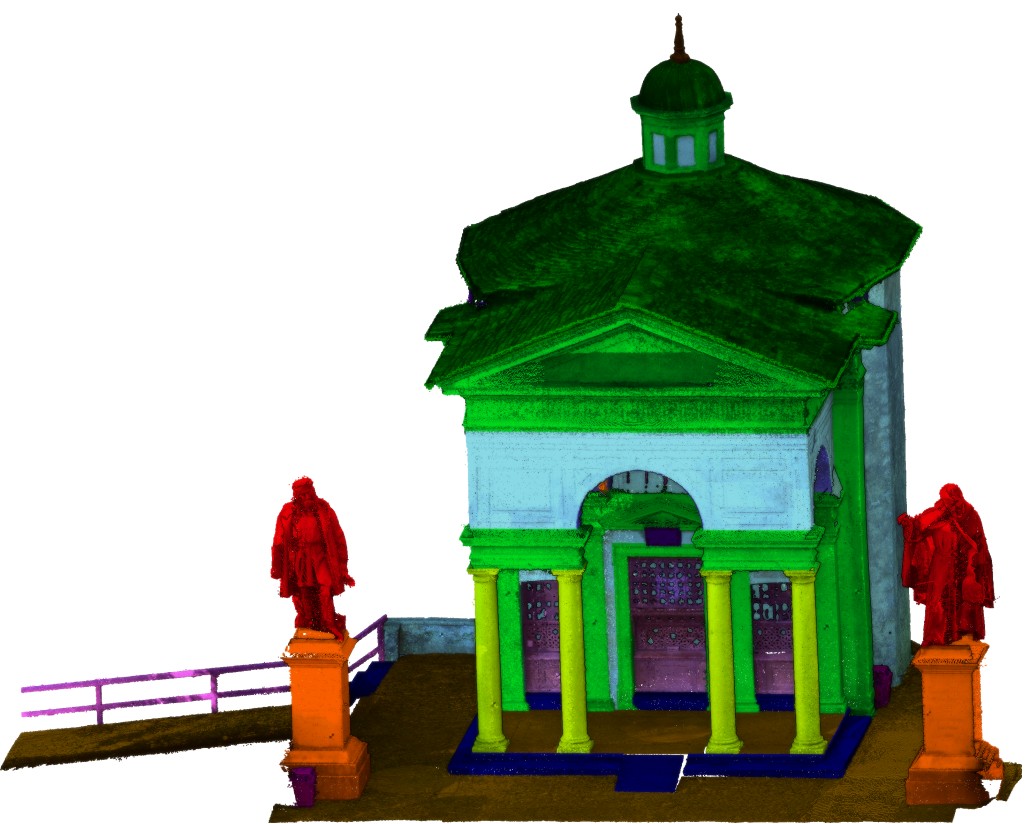

**Figure 17.** Optimal result obtained for the object recognition of the Sacro Monte chapel.

### 5.3. Object Enrichment Discussion

Results of object enrichment from Linked Open Data showed the addition of a range of knowledge and information related to recognized cultural objects. The knowledge is presented as text according to the RDF properties of the cultural object contained in the knowledge base, and the linked information is provided in different forms, such as texts, images, or websites. The use of Wikidata as the primary source of knowledge to enrich the knowledge base allows the process to retrieve and gather a large variety of information sources. Other knowledge sources linked to Wikidata and specialized in cultural heritage, such as Europeana [42], are easily integrated and exploitable thanks to the common use of CIDOC-CRM vocabulary. The enrichment process uses and thus benefits from the potential of both semantics and Open Data linked to Wikidata. In future works, it would be interesting to study the quantity and quality of the enriched links, as well as the quantity and types of knowledge and information sources integrated thanks to this process. The web interface could also be improved by functionalities, such as enrichment source selection and the search for specific knowledge or information related to a cultural object.

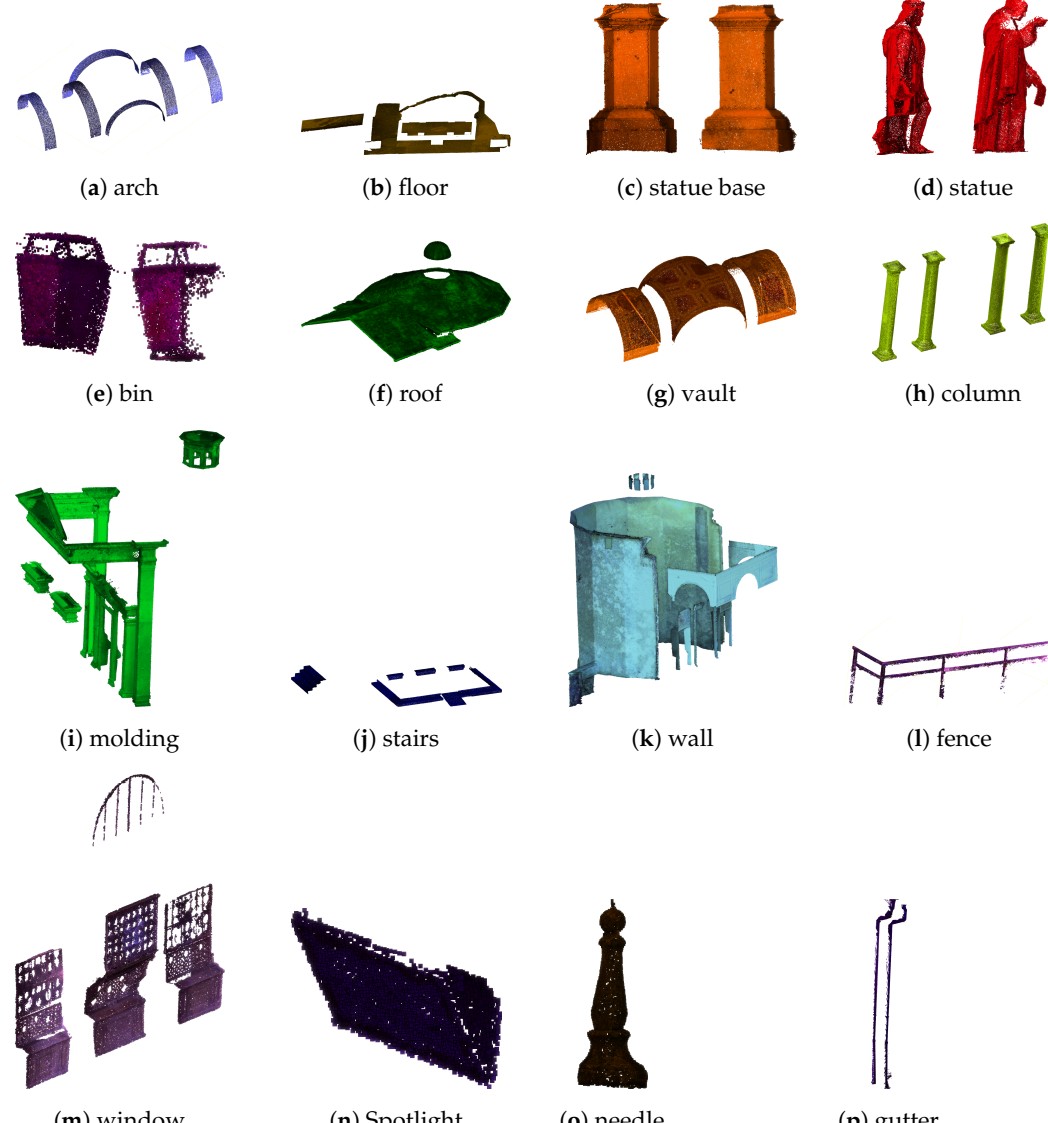

**Figure 18.** Isolated view of object recognized in the first chapel of Sacro Monte.

## 6. Conclusions

Semantic web technologies and Linked Open Data are increasingly utilized for the publishing of cultural heritage documentation. Documentation publishing, such as Linked Open Data, provides an essential source of knowledge and information that can be used to enrich and support cultural heritage object documentation. However, using Linked Open Data as an information source for cultural documentation requires the gathering of different sources from Linked Open Data to enrich the cultural documentation. Existing approaches using Linked Open Data as an information source for documentation are generally specific to a domain and focus on specific Linked Open Data sources. These approaches show semantic potential, but they do not entirely exploit this potential to support the documentation process from the acquisition to its presentation. Semantics can gather different knowledge domains, guide the documentation process in different contexts, and gather Linked Open Data sources for documentation enrichment with the goal of providing rich cultural heritage documentation. This study shows the semantic potential of these two approaches to support the end-to-end documentation process from data acquisition to cultural heritage presentation. The proposed method comprises three knowledge-based processing steps: acquisition technology recommendation, object recognition to structure the data, and data enrichment through Linked Open Data. This method

provides two main contributions. The first one is an end-to-end process to support the safeguard of cultural heritage. This end-to-end process is based on acquisition technology recommendations and object recognition, which can adapt to different contexts of cultural heritage. Thanks to this flexibility, the proposed method can support data digitization in its application to various cultural heritage cases. As shown through the two case studies, the proposed method is applicable to large cultural heritage objects, such as a terrace house, a watermill, or a chapel, but also smaller objects, such as statues. The second contribution is the gathering and centralization of a variety of information and documents related to cultural heritage objects, thanks to Linked Open Data. The flexibility and the connection between the different steps of the proposed methods are provided thanks to the semantics.

**Author Contributions:** Conceptualization, J.-J.P., C.P., and F.B.; methodology, J.-J.P., C.P., and F.B.; software, J.-J.P. and C.P.; validation, J.-J.P., C.P., and F.B.; formal analysis, J.-J.P., C.P., and F.B.; investigation, J.-J.P., C.P., and F.B.; resources, J.-J.P. and C.P.; data curation, J.-J.P. and C.P.; writing—original draft preparation, J.-J.P. and C.P.; writing—review and editing, J.-J.P., C.P., and F.B.; visualization, J.-J.P. and C.P.; supervision, F.B.; project administration, F.B. All authors have read and agreed to the published version of the manuscript.

**Funding:** This research received no external funding.

**Institutional Review Board Statement:** Not applicable.

**Informed Consent Statement:** Not applicable.

**Data Availability Statement:** The data supporting the reported results can be found at the following link: https://www.flyvast.com/flyvast/app/page-snapshot-viewer.html#/381/2cc70606-6227-7cdb-6de0-3c9e156d03c6, accessed on 7 May 2021.

**Acknowledgments:** The authors would like to thank the Austrian Archaeological Institute (ÖAI) and the Römisch-Germanisches Zentralmuseum (RGZM) for the permission to use the Ephesos point cloud.

**Conflicts of Interest:** The authors declare no conflict of interest.

## Abbreviations

The following abbreviations are used in this manuscript:

| | |
|---|---|
| KnowDIP | Knowledge-based Detection in Image and Point cloud |
| COSCH | Colour and Space in Cultural Heritage |
| COSCH-KR | COSCH Knowledge Representation |
| TLS | Terrestrial Laser Scanner |
| CH | Cultural Heritage |
| BIM | Building Information Modeling |
| HBIM | Heritage Building Information Modeling |
| RDF | Resource Description Framework |
| OWL | Web Ontology Language |
| CIDOC-CRM | Conceptual Reference Model |
| CNN | convolutional neural network |

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
