# Peer review of "From Acquisition to Presentation—The Potential of Semantics to Support the Safeguard of Cultural Heritage"

_remotesensing, doi:10.3390/rs13112226_

Round 1
Reviewer 1 Report
FROM ACQUISITION TO PRESENTATION - THE POTENTIAL OF SEMANTICS TO SUPPORT THE SAFEGUARD OF CULTURAL HERITAGE
Manuscript 1170391
The manuscript refers a process, executed on two Cultural Heritage constructions, to digitize, process the geometry/topology data, and add context/historical information to them. The authors have extensively applied techniques for knowledge administration and application for supporting this process.
ASSUMPTIONS FOR THE REVIEW
1. The Reviewer evaluates the manuscript under the RESEARCH ARTICLE category.
2. The Reviewer considers that the central obligation of the whole publication process is the benefit of the reader. All other considerations are subsidiary to the obligation to the reader. The reader must be informed in truthful, efficient, amicable manner, of the values of the manuscript and the manuscript must have very high editorial quality. This quality is needed to deliver the material to re reader in the easiest and most efficient possible manner.
3. The Reviewer has used the domains declared by (a) the Journal Remote Sensing , (b) the Special Issue "Remote Sensing, Metric Survey and Spatial Information Technologies for Heritage Management", as scenarios for the evaluation of the manuscript.
OVERALL APPRAISAL OF THE MANUSCRIPT
The Reviewer finds that the manuscript does not correspond to the domains (a) and (b) previously mentioned. By a decision of the authors, the manuscript concentrates its technical background discussion on the question of knowledge modeling, storage, administration and application. The manuscript does not claim to make a contribution on the domains (a) and (b) mentioned above. A contribution is a novel, pertinent, and relevant enlargement of the state-of-art , in the particular domain chosen. For a contribution to be present, it is not enough that the manuscript narrates an application on Cultural Heritage preservation. The contribution must be executed in the intersection of domains (a) and (b).
The reviewed manuscript does not have the structure of a research article. The manuscript is a Project Report. An important deficiency of the manuscript is that if fails to execute a Literature Review, IN THE PARTICULAR FIELD OF THE INTENDED CONTRIBUTION. The Reviewer considers that such a field must be in the intersection of (a) and (b) above. In addition, the Literature Review presents no structure or taxonomy on which the reader may rely. On the contrary, this manuscript section is an endless stream of acronyms and capitalizations, on the domain of Knowledge Engineering and in particular Ontologies, with no organization that helps the reader to relate it to the topics (a) and (b) above.
DECISION
Considering the emphasis and structure of the manuscript, the Reviewer considers that the most sensible course of action would be to advise the authors: (1) to submit their manuscript to a Knowledge Engineering Journal, and (b) to write a research article for such a purpose.
SPECIFIC COMMENTS.
1. In the Introduction, the manuscript presents a history of administrative, budgetary decisions of the European Union in the domain of Preservation of Cultural Heritage. It is, of course positive that such budgetary - administrative decisions have been made. However, their enumeration lies more in a Project Proposal or Report than in a Research Manuscript.
2. The authors claim that the knowledge modeling executed via Ontologies enables the correct decision of which sampling technologies to apply to a particular heritage object. However, the two examples presented correspond to the same type of (architectural) objects. The natural choice to prove the concept of SW-driven data collection and processing would be to have data variety (buildings, tools, weapons, kitchenware, arts), and many other object types related to human activities across the time.
3. It is not clear what the advantage is, of letting the Ontology to chose a sampling method for a particular object, vs. letting the architect or technician to do so (which, the manuscript itself declares to be the case in the Results section).
4. The topics of the special issue include but are not limited to:
4.1-3D documentation, modeling, and/or visualization in CH;
4.2-applications of drones in CH;
4.3-photogrammetry, laser scanning, and mobile/handheld data collection in CH;
4.4-heritage building information modeling (HBIM);
4.5-virtual, augmented, and mixed reality (VR/AR/MR) in CH;
4.6-data and sensor integration in CH;
4.7-archive data digitization in CH;
4.8-accuracy and precision assessment of CH modeling;
4.9-machine/deep learning in CH;
4.10-multi/hyperspectral analysis in CH;
4.11-restoration/rehabilitation in CH;
4.12-3D prototyping in CH;
4.13-underwater CH documentation; and
4.14-semantic classification of point clouds in CH.
from which 4.4 and 4.14 could be related to the present manuscript. The problem is that the authors have presented a manuscript whose theoretical and practical aspects are devoted to the problem of Knowledge Modeling and only tangentially to the problem of Remote Sensing for Cultural Heritage. The manuscript contributions in the area of Remote Sensing and Cultural Heritage are not clear. Its contributions in the domain of Knowledge Modeling and Administration would be, of course, more fully appreciated in a Journal of this domain. However, the Reviewer dares to express that, in such an hypothetical journal, the same observation would be made: (a) this is technical report and not a research article, (b) the research contribution or novelty in the domain of Knowledge Modeling is not clear.
5- Line 53: possible scenarios are: buildings, artifacts, artistic objects, frescos, paintings, mosaics, etc. However, the reported examples only address buildings.
6- Lines 87-91: These lines constitute, it seems, the proposed manuscript contribution. However, the specification is vague. It is not clear what the contribution in the area of Remote Sensing for Cultural Heritage is. "Contribution" is not the same as "application".
7- Lines 92-211: The Lit. Review orientates towards Knowledge Engineering, Ontologies. if the contribution of the manuscript is in this area, the manuscript belongs to a Knowledge Engineering Journal. If the contribution is in the area of Remote Sensing for Cultural Heritage, what is the contribution?
8- Lines 92-211: This review is basically unreadable. The authors transfer to the reader the problem of understanding the review and relief themselves of writing it for the reader benefit. In reality, of course, should be the authors’ problem. On the other hand, as a review, it is a sequence of comments without an organization or taxonomy. This lack of structure fits with the absence of clarity about the contribution of the paper.
9- Lines 199-211: The authors themselves bring their article outside the domain of Remote Sensing, Special Issue “"Remote Sensing, Metric Survey and Spatial Information Technologies for Heritage Management"”.
10- Line 217. The term "Objects" is very imprecise. Do the authors mean "features"? Are they addressing Level-of-Detail extraction and administration? it is not clear how the Ontologies handle Level-of-Detail. Some parts of the manuscript suggest automatic recognition of features. Others pin assign job to the reader.
11- Line 6, 222, and others: "end-to-end" mentions: The “end-to-end” process seems to be an important contribution. However, there is no formal definition of what this type of process it. In addition, is not every process end-to-end, by definition ? All depends on how the ends are defined. In addition: the manuscript itself dismisses its end-to-end claim: the samplings presented in the Cases of Study do not follow the Ontologies for Sampling, Acquisition and Processing that the manuscript itself proposes.
12- Liens 274-280: It seems that 1,2,3 can be reduced to 2. (1) does not exist, while (2) is part of (2). The three items are not clear. It is not clear what and "optimal input" would be.
13- Line 285. A "static algorithm" may be a common term in Knowledge Modeling. In computing languages it has another meaning. The issued is not a punctual one: is shows that the terminology of the manuscript aims to a readership not in the Remote Sensors Journal.
14- Lines 294-296: Do the authors mean Objects found in a site: tools, pots, weapons, houseware, etc. ? Or Do they mean “Features” (columns, frises, stairs, lobby, baluster, frieze, capital, etc.? There is a whole area of Geometric Modelinng, called Feature Recognition and Extraction. It his what the authors mean? How do THEIR Ontologies attack the problem of Feature Recognition ? What Features do they extract? what is the topological description of the Feature? Is there a Geometrical reduction of the topological Domain? What is the performance (complexity) of their Feature Extraction Algorithm? ¿How does Ontologies accelerate the process of Feature Extraction?
15- LINES: 298-307. What do Authors mean by “processing”? In shape reconstruction from point samples, typical processing includes: outlier detection and elimination, registration, identification and correction of high frequency neighborhoods, enforcement of manifold conditions, meshing, re-meshing, decimation, parameterization, control of non-manifold parameterizations, promotion to full Boundary Representation, Export to Special-app sw (e.g. structural mechanics, finite element analysis, manufacturing process planning, etc. ). Where is this process executed? Who executes it? This is a process that requires large amounts of human input, and several iterations. Do they mean the Ontologies execute this process?
16- Lines 322-335: It is not clear what the function owl:sameAs is and it is not clear what the object owl would be. As a matter of fact, the whole informatics discussion is carried out in the conceptual domain and not on the application domain. This characteristic reinforces the notion that the manuscript is written for a public in the Knowledge Modeling community and not in the Remote Sensing community.
17- Lines 336-343: Fig 1, Fig 3 and these lines exemplify why this manuscript is not for this journal: In real life, the gap between Panels A and B of Figure 3, corresponds to a very difficult and iterative process: identification an isolation of high frequency neighborhoods, data sampling, repair of self-occlusions, registration, data cleaning, inference of connectivity, correction of non-manifold conditions, re-meshing, decimation, parameterization, new correction of non-manifold conditions, promotion to Full Boundary Representation, export to specific application software (e.g. Comp. Mechanics, Finite Element Analysis, Geographic Information Systems, Arch Design, Construction, documentation, etc.). This process is intensive in well trained technical personnel, and it is iterative: last process failures may imply the need for physical re-sample, with all ensuing processes being repeated. The present manuscript expresses that all those process can be replaced by Ontologies and Knowledge Administration software. The Reviewer is an expert in this area, and is in the position of assuring that such statement seriously misleads the reader. It may be that the mislead is non-intentional. But is it present and unacceptable.
18- Lines 361-368: “Object Recognition” may have many meanings. ¿ Does it mean to recognize e.g. watermills ? What is the effectiveness of the Objet Recognition process? The authors claim it is, because is one of their achievements. And yet, it would be more appropriate for a Knowledge-oriented journal to screen the effectiveness of this identification. In any case, no data is given in this manuscript to asseess the effectiveness of the Object Recognition process.
19- Section 3.1.2. The history of the building is not an interest of the manuscript by itself. It is a very important and cherished landmark for a community, and worth of respect. However, the emphasis of the manuscript should be in the Remote Sensing of Cultural heritage and not in the history of the building in particular. The Reviewer remarks that the manuscript is extremely very long for a research article.
20- Line 392: RDF: Resource Description Framework. It means that the data has been processed already. Therefore, it is not clear why the manuscript claims to use Ontologies and Knowledge Administration for the Sampling and Processing.
21- Section 4 Results. The manuscript claims the application of ontologies for object recognition. Yet, it is the user who is (reportedly) executing these tasks. This inconsistency misleads the reader and explains the reason for the Reviewer to suggest the presentation of the manuscript to a Knowledge Engineering Journal.
22- Section 3.1.1 , line 355: If the physical sampling of the site was executed in 2009 / 2010, it is not clear how the Ontologies for determining the scanning equipment, settings and scan process itself were applied. The Reviewer could not find in the references indices of such application of Ontologies being the case in this data set. However, the Reviewer may be mistaken. In any case, a frontal clarification is needed on what the manuscript does NOT report and what it does INDEED report.
For example: the manuscript claims to appli ontologies for object equipment choice, sampling settings and processing settings. However, the 2 examples included seem to indicate that the data was already sampled and geometrically/topologically processed (e.g. lines 392, 355) when entering the system.
This reasoning would imply that the system is not an end-to-end one. Which follows that the claim of contribution of the manuscript is also not clear.
Author Response
Dear Reviewer,
Thank you very much for your comments, which helped us to improve the article. Please find below the answer to your questions and comments.
Best regards.
“The reviewed manuscript does not have the structure of a research article. The manuscript is a Project Report. An important deficiency of the manuscript is that if fails to execute a Literature Review, IN THE PARTICULAR FIELD OF THE INTENDED CONTRIBUTION. The Reviewer
considers that such a field must be in the intersection of (a) and (b) above. In addition, the Literature Review presents no structure or taxonomy on which the reader may rely. On the contrary, this manuscript section is an endless stream of acronyms and capitalizations, on the domain of Knowledge Engineering and in particular Ontologies, with no organization that helps the reader to relate it to the topics (a) and (b) above."
-> Concerning the structure of the article that you suggest modifying, the other readers do not seem to have the same opinion and point out that it is an interesting contribution to the cultural heritage protection communities and remote sensing. Moreover, the current structure follows the journal guidelines. Therefore, we do not have modified the article's structure to avoid disappointing the four other reviewers and to be sure to follow the journal guideline.
“1. In the Introduction, the manuscript presents a history of administrative, budgetary decisions of the European Union in the domain of Preservation of Cultural Heritage. It is, of course positive that such budgetary - administrative decisions have been made. However, their enumeration lies more in a Project Proposal or Report than in a Research Manuscript.”
-> Such information aims at showing the interest and the relevance of the presented research.
“2. The authors claim that the knowledge modeling executed via Ontologies enables the correct decision of which sampling technologies to apply to a particular heritage object. However, the two examples presented correspond to the same type of (architectural) objects. The natural choice to prove the concept of SW-driven data collection and processing would be to have data variety (buildings, tools, weapons, kitchenware, arts), and many other object types related to human activities across the time.”
-> From the data acquisition point of view, we agree with you that another example would be appreciated. However, from the object recognition point of view, such objects would not have much interest. That is why we have chosen these two examples, which are different from an archaeological point of view and with different objects inside, such as statues, moldings for the chapel.
“3. It is not clear what the advantage is, of letting the Ontology to chose a sampling method for a particular object, vs. letting the architect or technician to do so (which, the manuscript itself declares to be the case in the Results section).”
-> The main advantage is to not depend on such experts by using a representation of their knowledge.
“4. The topics of the special issue include but are not limited to:from which 4.4 and 4.14 could be related to the present manuscript. The problem is that the authors have presented a manuscript whose theoretical and practical aspects are devoted to the problem of Knowledge Modeling and
only tangentially to the problem of Remote Sensing for Cultural Heritage. The manuscript contributions in the area of Remote Sensing and Cultural Heritage are not clear. Its contributions in the domain of Knowledge Modeling and Administration would be, of course, more fully appreciated in a Journal of this domain. However, the Reviewer dares to express that, in such an hypothetical journal, the same observation would be made: (a) this is technical report and not a research article, (b) the research contribution or novelty in the domain of Knowledge Modeling is not clear.”
-> Thank you for your comment. We address the topics of 4.1-3D documentation, modeling, and/
or visualization in CH. The addressed problem is not the knowledge modeling but the overall cultural heritage documentation from digitized objects and buildings.
“5. Line 5: possible scenarios are: buildings, artifacts, artistic objects, frescos, paintings, mosaics, etc. However, the reported examples only address buildings. “
-> The chosen case studies are not limited to buildings. They also include artefacts and artistic objects. For example, the case of the Sacro monte chapel includes statues, moldings, columns, and vaults that contain paintings.
“6. Lines 87-91: These lines constitute, it seems, the proposed manuscript contribution. However, the specification is vague. It is not clear what the contribution in the area of Remote Sensing for Cultural Heritage is. "Contribution" is not the same as “application"."
“7- Lines 92-211: The Lit. Review orientates towards Knowledge Engineering, Ontologies. if the contribution of the manuscript is in this area, the manuscript belongs to a Knowledge Engineering Journal. If the contribution is in the area of Remote Sensing for Cultural Heritage, what is the contribution?”
-> Contributions are explicitly stated in the conclusion section (lines 635 to 645): « This method provides two main contributions. The first one is an end-to-end process to support the cultural heritage safeguard. This end-to-end process is based on acquisition technologies recommendation and object recognition, which can adapt to different contexts of cultural heritage. Thanks to this flexibility, the proposed method can support data digitization to its presentation for various cultural heritage application cases. As it has been shown through the two case studies, the proposed method is applicable to large cultural heritage objects as a terrace house, a watermill, or a chapel, but also smaller objects as statues. The second contribution is to gather and centralize a diversity of information and documents related to cultural heritage objects, thanks to Linked Open Data. The flexibility and the connection between the different steps of
the proposed methods are provided thanks to the semantics. »
“8- Lines 92-211: This review is basically unreadable. The authors transfer to the reader the problem of understanding the review and relief themselves of writing it for the reader benefit. In reality, of course, should be the authors’ problem. On the other hand, as a review, it is a sequence of comments without an organization or taxonomy. This lack of structure fits with the absence of clarity about the contribution of the paper. ”
-> We have improved the review section to allow a better understanding.
“9- Lines 199-211: The authors themselves bring their article outside the domain of Remote Sensing, Special Issue “"Remote Sensing, Metric Survey and Spatial Information Technologies for Heritage Management"”.”
-> We have introduced the method context and provided an overview of our approach.
“10- Line 217. The term "Objects" is very imprecise. Do the authors mean "features"? Are they addressing Level-of-Detail extraction and administration? it is not clear how the Ontologies handle Level-of-Detail. Some parts of the manuscript suggest automatic recognition of features. Others pin assign job to the reader. ”
-> By "objects", we mean archaeological objects (e.g. chapel, watermill, statue…) or parts of archaeological/structural object (e.g. moldings, arch, …). We do not address Level-of-Detail extraction and administration. We automatically identify features to recognize objects and elements semantically.
“11- Line 6, 222, and others: "end-to-end" mentions: The “end-to-end” process seems to be an important contribution. However, there is no formal definition of what this type of process it. In addition, is not every process end-to-end, by definition ? All depends on how the ends are defined. In addition: the manuscript itself dismisses its end-to-end claim: the samplings presented in the Cases of Study do not follow the Ontologies for Sampling, Acquisition and Processing that the manuscript itself proposes. ”
-> Processes are not end-to-end when they required human interaction. The sampling provided in the case study section follows the presented ontology by using classes presented in the Method and represent only the user input.
“12- Liens 274-280: It seems that 1,2,3 can be reduced to 2. (1) does not exist, while (2) is part of (2). The three items are not clear. It is not clear what and "optimal input" would be. ”
-> We do not well understand this comment. Although the three steps are related to each other, they require independent processing steps. Our approach is composed of these three steps.
Optimal input is denoised data with enough accuracy according to the application case (object to be digitized).
“13- Line 285. A "static algorithm" may be a common term in Knowledge Modeling. In computing languages it has another meaning. The issued is not a punctual one: is shows that the terminology of the manuscript aims to a readership not in the Remote Sensors Journal.”
-> We have added an explanation of this terminology.
“14- Lines 294-296: Do the authors mean Objects found in a site: tools, pots, weapons, houseware, etc. ? Or Do they mean “Features” (columns, frises, stairs, lobby, baluster, frieze, capital, etc.? There is a whole area of Geometric Modelinng, called Feature Recognition and Extraction. It his what the authors mean? How do THEIR Ontologies attack the problem of Feature Recognition ? What Features do they extract? what is the topological description of the Feature? Is there a Geometrical reduction of the topological Domain? What is the performance (complexity) of their Feature Extraction Algorithm? ¿How does Ontologies accelerate the process of Feature Extraction?”
-> The ontology guides the features extraction process that describes objects (columns, frises,
stairs, lobby, baluster, frieze, capital) and not aims to accelerate the process. The complete explanation of the working process behind is explained in works that are referenced in the article (27,36,37,41,42,43,45)
“15- LINES: 298-307. What do Authors mean by “processing”? In shape reconstruction from point samples, typical processing includes: outlier detection and elimination, registration, identification and correction of high frequency neighborhoods, enforcement of manifold conditions, meshing,
re-meshing, decimation, parameterization, control of non-manifold parameterizations, promotion to full Boundary Representation, Export to Special-app sw (e.g. structural mechanics, finite element analysis, manufacturing process planning, etc. ). Where is this process executed? Who executes it? This is a process that requires large amounts of human input, and several iterations. Do they mean the Ontologies execute this process?”
-> By processing, we mean data processing for object recognition and not for shape reconstruction. The ontology executes this process according to the input presented in section 4.1. Recommendation of acquisition technologies
Remote sensing community include other community as Machine learning, deep learning, Knowledge Modeling, etc.
“16- Lines 322-335: It is not clear what the function owl:sameAs is and it is not clear what the object owl would be. As a matter of fact, the whole informatics discussion is carried out in the conceptual domain and not on the application domain. This characteristic reinforces the notion that the manuscript is written for a public in the Knowledge Modeling community and not in the Remote Sensing community. ”
-> We have added explanations of these meaning for non-experts in Semantics.
“17- Lines 336-343: Fig 1, Fig 3 and these lines exemplify why this manuscript is not for this journal: In real life, the gap between Panels A and B of Figure 3, corresponds to a very difficult and iterative process: identification an isolation of high frequency neighborhoods, data sampling, repair of self-occlusions, registration, data cleaning, inference of connectivity, correction of non-manifold conditions, re-meshing, decimation, parameterization, new correction of non-manifold conditions, promotion to Full Boundary Representation, export to specific application software (e.g. Comp.
Mechanics, Finite Element Analysis, Geographic Information Systems, Arch Design, Construction, documentation, etc.). This process is intensive in well trained technical personnel, and it is iterative: last process failures may imply the need for physical re-sample, with all ensuing processes being repeated. The present manuscript expresses that all those process can be replaced by Ontologies and Knowledge Administration software. The Reviewer is an expert in this area, and is in the position of assuring that such statement seriously misleads the reader. It may be that the mislead is non-intentional. But is it present and unacceptable.”
-> We recommend that the reviewer looks at the following references (27,36,37,41,42,43,45) for
more details about the object recognition process and data acquisition recommendation. As the reviewer state that the manuscript is long, we would not extend it by adding details that refer to published research.
“18- Lines 361-368: “Object Recognition” may have many meanings. ¿ Does it mean to recognize
e.g. watermills ? What is the effectiveness of the Objet Recognition process? The authors claim it is, because is one of their achievements. And yet, it would be more appropriate for a Knowledge- oriented journal to screen the effectiveness of this identification. In any case, no data is given in this manuscript to asseess the effectiveness of the Object Recognition process.”
-> We do not understand what the reviewer means. The meaning, results and assessment of
object recognition is presented in section 4.2. Object recognition results. Moreover, our data are provided in the manuscript in the section Data Availability Statement.
“19- Section 3.1.2. The history of the building is not an interest of the manuscript by itself. It is a very important and cherished landmark for a community, and worth of respect. However, the emphasis of the manuscript should be in the Remote Sensing of Cultural heritage and not in the history of the building in particular. The Reviewer remarks that the manuscript is extremely very long for a research article. ”
-> The historical aspect helps the data processing by providing clues to the object localization, as explained in both use cases.
“20- Line 392: RDF: Resource Description Framework. It means that the data has been processed already. Therefore, it is not clear why the manuscript claims to use Ontologies and Knowledge Administration for the Sampling and Processing.”
-> We refer to the definition of RDF that is an essential component of semantic web technologies. It does not mean that the data has been processed already.
“21- Section 4 Results. The manuscript claims the application of ontologies for object recognition. Yet, it is the user who is (reportedly) executing these tasks. This inconsistency misleads the reader and explains the reason for the Reviewer to suggest the presentation of the manuscript to a Knowledge Engineering Journal.”
-> The user provides data after the proposed method recommends the technology to use, but the data processing and object recognition are fully automated.
“22- Section 3.1.1 , line 355: If the physical sampling of the site was executed in 2009 / 2010, it is not clear how the Ontologies for determining the scanning equipment, settings and scan process itself were applied. The Reviewer could not find in the references indices of such application of Ontologies being the case in this data set. However, the Reviewer may be mistaken. In any case, a frontal clarification is needed on what the manuscript does NOT report and what it does INDEED report. ”
-> The data are not processed before entering the system. The system provides such a process automatically, as described in the Method section.
Reviewer 2 Report
This is an excellent and original article that is very helpful to those working in the cultural heritage field. It extends current work on linked open data and adds valuable information to the field.
Author Response
Dear Reviewer,
Thank you very much for your comments. Best regards.
Reviewer 3 Report
Well-described article. I would like to see this paper published after considering the comments mentioned below.
The introduction provides sufficient background, however, it would be beneficial if the authors may refer to some archaeological works based on remote sensing datasets acquired in challenging (underwater) environments, like lakes, and seas. For example, consider the very recent works of:
- Yang, F.; Xu, F.; Zhang, K.; Bu, X.; Hu, H.; Anokye, M. Characterisation of Terrain Variations of an Underwater Ancient Town in Qiandao Lake. Remote Sensing 2020, 12, doi:10.3390/rs12020268.
- Janowski, L.; Kubacka, M.; Pydyn, A.; Popek, M.; Gajewski, L. From acoustics to underwater archaeology: Deep investigation of a shallow lake using high‐resolution hydroacoustics—The case of Lake Lednica, Poland. Archaeometry 2021, 10.1111/arcm.12663, doi:10.1111/arcm.12663.
- Majcher, J.; Quinn, R.; Plets, R.; Coughlan, M.; McGonigle, C.; Sacchetti, F.; Westley, K. Spatial and temporal variability in geomorphic change at tidally influenced shipwreck sites: The use of time‐lapse multibeam data for the assessment of site formation processes. Geoarchaeology 2021, 10.1002/gea.21840, doi:10.1002/gea.21840.
lines 357-358; 388-389: please, provide the exact device and its manufacturer used for data acquisition. Also, provide the processing description with the software used for data processing and cleaning.
lines 452-454: did you use any automatic way of object recognition or it was done just manually?
lines 471-473: what was the basis for the segmentation process? Did you use any automatic segmentation procedure? Please clarify.
line 476: Vague. Provide a list of used data processing algorithms.
line 496: Vague. Provide a methodology basis for accuracy assessment calculations.
lines 554-556: Both measurement methods have their own (qualitative) precision and accuracy. You do not have to deduce.
lines 618-645: formatting issues
Author Response
Dear Reviewer,
Thank you very much for your comments, which helped us to improve the article. Please find below the answer to your questions and comments.
Best regards.
“The introduction provides sufficient background, however, it would be beneficial if the authors may refer to some archaeological works based on remote sensing datasets acquired in challenging (underwater) environments, like lakes, and seas. For example, consider the very recent works of:
- Yang, ; Xu, F.; Zhang, K.; Bu, X.; Hu, H.; Anokye, M. Characterisation of Terrain Variations of an Underwater Ancient Town in Qiandao Lake. Remote Sensing 2020, 12, doi:10.3390/rs12020268.
- Janowski, L.; Kubacka, ; Pydyn, A.; Popek, M.; Gajewski, L. From acoustics to underwater archaeology: Deep investigation of a shallow lake using high‐resolution hydroacoustics—The case of Lake Lednica, Poland. Archaeometry 2021, 10.1111/arcm.12663, doi:10.1111/arcm.12663.
- Majcher, J.; Quinn, R.; Plets, R.; Coughlan, M.; McGonigle, C.; Sacchetti, F.; Westley, K. Spatial and temporal variability in geomorphic change at tidally influenced shipwreck sites: The use of time‐lapse multibeam data for the assessment of site formation Geoarchaeology 2021, 10.1002/gea.21840, doi:10.1002/gea.21840.”
-> Thank you for this comment and the references. We have added them in the section Introduction.
“lines 357-358; 388-389: please, provide the exact device and its manufacturer used for data acquisition. Also, provide the processing description with the software used for data processing and cleaning”
-> We have added the exact devices used for data acquisition and provided information that we have about the data processing (c.f. the enclosed track change file).
“lines 452-454: did you use any automatic way of object recognition or it was done just manually?”
-> It is fully automated. We have added precisions in these lines to clarify it.
“lines 471-473: what was the basis for the segmentation process? Did you use any automatic segmentation procedure? Please clarify.”
-> We have clarified it by specifying in the paper the type of segmentation used.
“line 476: Vague. Provide a list of used data processing algorithms.”
-> The used data processing algorithms have been added to the paper.
“line 496: Vague. Provide a methodology basis for accuracy assessment calculations.”
-> We do not well understand your comment. The Precision, F1-score and Recall are often used methodology for accuracy assessments. Do we need to describe the calculations of these metrics?
“lines 554-556: Both measurement methods have their own (qualitative) precision and accuracy. You do not have to deduce.”
-> We have removed this sentence.
“lines 618-645: formatting issues”
-> We have fixed these formatting issues.
Reviewer 4 Report
As the authors state, the 3D scanning process plays an important role in both the conservation and research of historical monuments and archaeological remains. In this sense, any methodology aimed at improving the usefulness of data acquisition and presentation should be welcomed. This paper offers an interesting end-to-end approach (from data acquisition to presentation) based on Semantic Web technologies that undoubtedly provides an essential source of knowledge.
Author Response

(The authors gave the same response as above.)

Reviewer 5 Report
The authors address a crucial problem in the investigation of cultural heritage objects, the selection of acquisition methods, data evaluation and data presentation in an interdisciplinary team. In worst case the person understanding the technical side does not understand the object and vice versa.
The authors propose an end-to-end process for acquisition, processing and presentation of 3D data, including linking it to existing data bases, making use of semantic enrichment and ontology. The workflow is demonstrated in two case studies: water mills in the ruins of Ephesos and a 16th century chapel.
While I am not qualified to judge the quality and practicality of their approach, I consider it from the perspective of heritage studies as novel and very promising and as such the manuscript deems publication.
The manuscript has, however, severe limits. While the research is clearly interdisciplinary the manuscript is solely aiming at a sub-part of the remote sensing community. It is in its current form not accessible to the desired “customers”, heritage professionals. I would suggest an introduction aimed at this field in that, e.g. semantics, are explained for scholars.
Further, it would be nice to add more explicitly which step in the work process was done by hand and how much time it took and how much it saved.
Author Response
Dear Reviewer,
Thank you very much for your comments, which helped us to improve the article. Please find below the answer to your questions and comments.
Best regards.
“I would suggest an introduction aimed at this field in that, e.g. semantics, are explained for scholars.”
-> We have added such an explanation into the introduction (c.f. the enclosed track change file).
“Further, it would be nice to add more explicitly which step in the work process was done by hand and how much time it took and how much it saved.”
-> We have highlighted processes done by hand, which correspond to the user input by adding a legend to figure 1 (Green arrows correspond to user inputs, and red arrows correspond to automatically generated outputs).
Round 2
Reviewer 1 Report
FROM ACQUISITION TO PRESENTATION - THE POTENTIAL OF SEMANTICS TO SUPPORT THE SAFEGUARD OF CULTURAL HERITAGE
Review for Version 2
The Reviewer has the following position regarding the Author Response to the 1st review:
1- The authors have neglected each one of the observations or petitions of the Reviewer.
2- The author responses to the observations of the reviewer confirm the Reviewer position: There is no clarity on the contributions of the manuscript. In the responses, the authors declare that their contribution is NOT in the Knowledge Modeling (KM) domain. However, the manuscript is in an overwhelming portion devoted to KM.
3- The References section show that all the aspects presented in the manuscript have been already published ny the authors. References 27, 36, 37, 41 42, 43, 45 already inform on
Knowledge-based object recognition in point clouds, Object detection in unstructured 3D data sets using explicit semantics, Automatic detection of objects in 3D point clouds based on exclusively semantic guided processes, Connected Semantic Concepts as a Base for Optimal Recording and Computer-Based Modelling of Cultural Heritage Objects, 3D object recognition through a process based on semantics and consideration of the context , Identification and classification of objects in 3D point clouds based on a semantic concept, Self-learning ontology for instance segmentation of 3D indoor point cloud.
, therefore, it is not clear what is the contribution of the authors in the present manuscript.
4- The originality of the methods mentioned in the Abstract and the Manuscript are refuted by the manuscript itself:
(1) The KM recommendation on Hardware, Setting and data collection is not used by the authors, because they use Open Data from other authors.
(2) Object recognition is not proven, because the authors do not show the RIGHT vs WRONG statistics of the recognition.
(3) The model enrichment process is being, at least, since 2003, 2004 (project GHEIST, Fraunhofer Inst. for Computer Graphics, Darmstadt Germany).
===========================================================
Regarding the claimed contributions:
(A) End-to-end processing. The manuscript has presented no discussion to document the end-to-end contribution: (a) the data collection is not exercised, as the samples come from Open Sets, (b) the automated choice of equipment and settings is not documented (and it seems to be treated in other publications by the authors' team).
(B) The enrichment of information has been, again, published by the authors in other articles. In such a sense, is not a contribution of the manuscript. As said before, this enrichment is by no means a novelty: The project GHEIST by Fraunhofer Inst. of Computer Graphics (Jasnoch & Krestshmer et al 2003, 2004,..) documents the recognition of the building by image processing techniques, the addition of historical background, the completion of remains by using a geometric model DataDase, in the context of Virtal Reality - aided Tourism.
The authors mention Linked Open Data as a software support. Therefore, the software implementation is not a contribution of the manuscript either.
When asked to document a contribution in the area of KM, the authors respond that this is not the area of the contribution, and that such topic is addressed in other publications. Yet, the manuscript is mostly devoted to Knowledge Modeling.
When asked to document the Feature Extraction (Object Recongnition) process, the authors respond by pointing to their previous publications. However, they present such process as a novelty in this manuscript.
Religious Bias:
The Reviewer is a person of catholic ancestry. However the Reviewer finds inconvenient the reference to catholic theological questions in the manuscript and considers that any manuscript in a scientific / technical journal has no place for religious discussions.
Break of Protocols in the Review process.
It is customary that the review process should not become an exchange between Author and Reviewer. Because of this reason:
(a) Except for cases of frontal contradiction with the Reviewer, all the responses of the Author must be in terms of CHANGES TO THE MANUSCRIPT, abiding by the observation of the Reviewer. In no case, a private argumentation Author VS Reviewer is used. Even if the Reviewer is wrong and mistaken, the Author (for the benefit of the reader) examines and corrects the manuscript, because the reader might be misled, as the Reviewer was.
(b) The Author is not entitled to reject a petition of the Reviewer on the basis that other Reviewers did not ask for it. This Reviewer has never witnessed in 30+ years of publishing , that a Reviewer or Editor penalizes the Author for abiding by the petitions of another Reviewer. This attempt to pit one Reviewer agains the other is completely unacceptable.
===========================================================
SUMMARY:
The work reported in the manuscript is an vast one. The Reviewer recognizes that. However, the needs when publishing a research article are: (a) the rights of the reader to an efficient, truthful and novel material must be respected. (b) the manuscript must make a strong case for novel material, (c) the material must be in agreement with the direction of the journal and the special issue or scenario that the publisher envisions.
In the present case, the Reviewer contents:
1- The authors failed to make the corrections and clarifications that the initial review diagnosed as needed. The Journal asked the Reviewer to evaluate this second manuscript. This means that the Journal considers the Reviewer's observations/petitions as administratively related to the article, in spite of the initial Reviewer recommendation of REJECTION.
2- The authors fail to present the case for novel material, in relation to the state-of-art in general, and the authors' previous publications, in particular.
3- The manuscript is composed as a Project Report and not as a Research Article.
4- The authors contradict themselves regarding the contributions of the manuscript. In the responses to 1st. review they declare that Knowledge Modeling is not a field of their contribution. However, they devote the full manuscript to Knowledge Modeling (and not Remote Sensors). Still, they cannot articulate the contribution of the manuscript.
5- The manuscript contains references to administrative, budgetary, religious matters, that are not the concern of a scientific / technical publication.
NOTE: The 2nd version contains ? symbols instead of the reference numbers. The references themselves are not numbered. The Reviewer turned to the 1st version for this aspect.
